

# Transport of large stratospheric ozone to the surface by a dying typhoon and shallow convection

Zhixiong Chen[1,3], Jane Liu[1,2], Xiushu Qie[3], Xugeng Cheng[1], Yukun Shen[1], Mengmiao Yang[1], Xiangke Liu[4]

[1] Key Laboratory for Humid Subtropical Eco-Geographical Processes of the Ministry of Education, School of Geographical Sciences, Fujian Normal University, Fuzhou, China
[2] Department of Geography and Planning, University of Toronto, Toronto, Ontario, Canada
[3] Key Laboratory of Middle Atmosphere and Global Environment Observation (LAGEO), Institute of Atmospheric Physics, Chinese Academy of Sciences, Beijing, China
[4] Key Laboratory for Meteorological Disaster Prevention and Mitigation of Shandong, Jinan, China

Correspondence: Jane Liu (janejj.liu@utoronto.ca)

**Abstract.** Stratospheric ozone transported to the troposphere is estimated to account for 5-10% of the tropospheric ozone sources. However, chances for intruded stratospheric ozone to reach the surface are low. Here, we report an event of strong surface ozone surge with stratospheric origins in the North China Plain (NCP, 34ºN-40ºN, 114ºE-121ºE) at night of 31 July 2021. The hourly measurements revealed that surface ozone concentrations were up to 80-90 ppbv at several cities over the NCP from 23:00 on 31 July 1 to 6:00 on 01 August, 2021, which was 40-50 ppbv higher than the corresponding monthly mean. A high-frequency surface measurement indicates that this ozone surge occurred abruptly and reached 40-50 ppbv within ~10 minutes. A concurrent decline in surface carbon monoxide (CO) concentrations suggests that this surface ozone surge resulted from downward transport of stratospheric ozone-rich and CO-poor airmass. This is further confirmed by the vertical evolutions of humidity and ozone profiles at night, based on radiosonde and satellite data, respectively. Such an event of stratospheric impact on surface ozone is rarely documented in terms of its magnitude, covering areas, abruptness, and duration.

We find that this surface ozone surge was induced by a combined effect of a dying typhoon In-fa and shallow local mesoscale convective systems (MCS) that facilitated the transport of stratospheric ozone to the surface. This finding is based on analysis of meteorological reanalysis and radiosonde data, combining with high-resolution FLEXPART-WRF modeling. (WRF: Weather Research and Forecasting, FLEXPART: Flexible Lagrangian particle dispersion model). Although the synoptic-scale typhoon In-fa was in dissipation stage when it passed through the NCP, it could still bring down stratospheric dry and ozone-rich airmass. As a result, the stratospheric airmass descended to the middle-to-low troposphere over the NCP before the MCS formed. With the pre-existed stratospheric airmass, the convective downdrafts of the MCS facilitated the final descending of stratospheric airmass to the surface. Significant surface ozone enhancement occurred in the convective downdraft regions during the development and propagation of the MCS. This study underscores the non-negligible roles of dying typhoons and shallow convection in the transport of stratospheric ozone to the troposphere and even the surface, which have important implications for air quality, tropospheric ozone budget, and climate change.



## 1 Introduction

The exchange between the stratosphere and troposphere, between which chemical compositions and static stability are
fundamentally different, is crucial to atmospheric chemistry, global climate change, and ecosystem health (Holton et al., 1995;
Stohl et al., 2003). The stratosphere stores approximately 90-95% of atmospheric ozone ($O_3$), and hence is characterized with
high abundance of ozone. Meanwhile, the stratosphere contains little water vapour and carbon monoxide (CO) that is primarily
emitted from combustion processes near the surface (Hartmann et al., 2001; Pan et al., 2014, 2018; Li, D. et al., 2020). In
contrast, the troposphere contains only 5-10 % of atmospheric ozone, and is with high water vapor and CO, due to its closeness
to the surface sources. Therefore, tropospheric airmass is relatively rich in CO and water vapor, and poor in ozone. The
exchange of these trace gases between the stratosphere and troposphere across the tropopause, a separation between the
stratosphere and troposphere, can occur under the influences of synoptic-scale and meso-scale atmospheric processes. Among
these processes, deep convection is of great interest because they can effectively redistribute the trace gases vertically by
modulating the flows of airmass upward or downward (Dickerson et al., 1987; Lelieveld and Crutzen, 1994; Pickering et al.,
1991, 1992; Li et al., 2017). For example, intensive updrafts of deep convection can transport ozone and its precursors like
CO, nitrogen oxides ($NO_x$) and volatile organic compounds (VOC) in the atmospheric boundary layer (ABL) to the upper
troposphere and lower stratosphere (UTLS), and hence alter the chemical nature and promote the substantial ozone formation
in UTLS. The stratospheric ozone-rich airmass can also be transported downward to the lower troposphere by deep convection.
Therefore, deep convection is deemed important to the ozone budgets in the stratosphere and troposphere.

Previous studies on convective redistribution of vertical atmospheric composition mainly focus on the upward injection of
pollutants from ABL to UTLS, while recent field campaigns and numerical analysis start to pay attention to the downward
transport of stratospheric airmass and its influences on the troposphere (e.g., Baray et al., 1999; Betts et al., 2002; Sahu and
Lal, 2006; Hu et al., 2010; Pan et al., 2014; Phoenix et al., 2020; 2021). It is known that ozone is important for radiation balance
of climate system and atmospheric oxidative capability. In recent years, continuous increases in surface ozone levels over
many areas in China were reported (Li et al., 2019; Han et al., 2020), while the contributions from the stratosphere to
troposphere (STT) exchange processes to the increasing surface ozone have been studied little. There are large uncertainties
in the estimation of stratospheric impacts on the tropospheric ozone budget, because most studies are based on global models
that are with coarse spatiotemporal resolutions and simplified representation of convection. Though events of stratospheric
intrusions directly influencing surface ozone concentrations appear rare and sporadic (Davies and Schuepbach, 1994), the
frequency and intensity of convection are projected to increase significantly in the future as a result of global warming (Del
Genio et al., 2007; Raupach et al. 2021), which raises the likelihood of enhanced convection-triggered STT exchanges in the
future. Therefore, detailed analysis of simulations with high spatiotemporal resolution models can enhance our understanding
of stratospheric intrusion related to convection.

The variation in ozone concentrations in the troposphere has close linkages with stratospheric intrusions of ozone-rich
airmass through convection. For example, Pan et al. (2014), based on aircraft observations, found that the stratospheric ozone-





rich airmass can be transported downward and wrapped around the anvil by mesoscale convective systems (MCS) with overshooting convection. Pan et al. (2014) and Phoenix et al. (2020) revealed that vigorous atmospheric motions of tropopause-penetrating convection can perturbate the tropopause and drive subsidence flow containing stratospheric ozone-rich airmass around the storm edges. Researchers also observed that small-scale convective downdrafts over tropical regions such as

Amazon rainforest are able to enhance surface ozone by 3-30 ppbv (Betts, et al. 2002; Grant et al. 2008; Gerken et al., 2016; Melo et al., 2019). Jiang et al. (2005) reported a typhoon-induced high ozone episode at night with large surface ozone increases reaching 21-42 ppbv over the southeastern coast of China. Along the downward transport of stratospheric ozone-rich airmass, the upper and middle troposphere are most frequently impacted by the intrusions that mix with ambient air and contribute to the general free tropospheric ozone burden (Zanis et al., 2003; Tarasick et al., 2019). In some cases, stratospheric airmass can

sink to the surface, while there are still some fundamental problems related to the deep stratospheric intrusions that require in-depth investigation. In this study, we report an event of vigorous surface ozone enhancement with stratospheric origins observed at midnight on 31 July 2021 over the North China Plain (NCP) (34-40 °N, 114-121 °E, geographical location is shown in Fig. 1). Impacted by the typhoon In-fa and local MCS, the stratospheric ozone-rich airmass was transported downward to the surface, and surface ozone concentrations reached 80-90 ppbv at several cities over the NCP from 23:00 on

31 July 1 to 6:00 on 01 August, 2021. Compare with the monthly mean ozone concentrations, surface ozone was enhanced by up to 40-50 ppbv. The surface ozone concentrations in this stratospheric intrusion event were, on average, two times larger than the normal nighttime level of ozone over the NCP. Such significant ozone surge is impressive, given the rareness of direct stratospheric intrusions into the ground level and severe threats to the ecosystem. In addition, several features of atmospheric processes responsible for this nighttime surface ozone surge event are worth noting. First, upon the occurrence of the ozone

surge, typhoon In-fa, which caused the record-breaking rainfall over Henan province of northern China in 2021 summer, had been downgraded to tropical depression (TD, with wind speed of 10.8-17.1 m s$^{-1}$) category and evolved into dissipation stage. Previous study evaluating the impacts of typhoons on tropospheric ozone shows that typhoons can induce stratospheric intrusions to the lower troposphere when they are intensive over the ocean (Chen et al. 2021). While in this case, typhoon In-fa had made landfall on 25 July 2021, and was very weak when it moved into the NCP on 29 July, but can still pose

unneglectable influences on the tropospheric ozone. Second, instead of showing significant tropopause-penetrating features in the convection case of Pan et al. (2014), the local MCS associated with the ozone surge were shallow in terms of vertical development and did not penetrate into the tropopause. Since there are few studies that documented and analysed the stratospheric impact on the troposphere over the NCP (Li et al., 2015a, b), the variations, magnitudes, transport pathways, and mechanisms of how stratospheric airmass can reach the surface remain less understood.

Specifically, how the stratospheric airmass finally descend to the ground level is not clear, despite some detrainment processes of stratospheric ozone to ambient air in the upper and middle troposphere. Therefore, based on the observations and model simulations with high spatiotemporal resolutions, we intend to address the following key scientific issues related to this surface ozone surge that is induced by stratospheric intrusions:



(1) The fine-scale spatiotemporal variations and magnitudes of surface ozone enhancement induced by the stratospheric
intrusions.

(2) The interactions between synoptic-scale and meso-scale atmospheric processes responsible for the rapid and direct
stratospheric influences.

(3) The transport pathways of stratospheric ozone-rich airmass to reach the surface.

The remaining paper is structured as follows. Section 2 describes the atmospheric composition observational data and
meteorological data. Details of high-resolution simulations of the MCS and backward trajectories analysis are also introduced.
Section 3 presents the fine-scale variations in surface atmospheric composition. In Section 4, we analyse the multi-scale
interactions of atmospheric processes responsible for the stratospheric intrusion to the surface, and present the transport
pathways of ozone-rich airmass. Section 5 offers the conclusions and discussions.

## 2 Data and Model

### 110    2.1 Atmospheric composition observations

Ground-based air pollutant data were collected from two sources. Firstly, a nationwide observation network with more
than 1500 stations distributed over 454 cities is maintained by China National Environmental Monitoring Centre (CNEMC),
which measures air pollutants including surface fine particles with an aerodynamic diameter of up to 2.5 μm ($PM_{2.5}$) and of up
to 10 μm ($PM_{10}$), $O_3$, CO, nitrogen dioxide ($NO_2$) and sulphur dioxide ($SO_2$) (Lu et al., 2018). The air pollutant observations
from CNEMC are strictly quality controlled and released with a 1-hour temporal resolution (http://106.37.208.233:20035/, last
access: 21 January 2022). Correspondingly, city-scale air pollutant concentrations were obtained by averaging all available
station observations in cities such as Hengshui (HS), Jinan (JN), Binzhou (BZ), Weifang (WF), Qingdao (QD) and Weihai
(WH) (Fig.1). Secondly, continuous measurements of $O_3$, CO, and $NO_x$ were made in July-August 2021 at a rural station
(37.82 °N, 118.11 °E) located in Zhanhua (ZH), a county of Binzhou city, where the field campaign of 2021 Shandong
Triggering Lightning Experiment (SHATLE) was performed by Institute of Atmospheric Physics (IAP) of Chinese Academy
of Sciences (CAS; Qie et al., 2009; Jiang et al., 2013). The applied atmospheric composition instruments include an ultraviolet
photometric $O_3$ analyzer (Model 49i), a $NO_x$ analyzer (Model 42i-TL) and a CO analyzer (Model 48i-TLE) produced by
Thermo Fisher Scientific Inc. Detailed calibrations and daily maintenance were performed to ensure data quality. $O_3$, CO, and
$NO_x$ concentrations (in ppbv) were output at a frequency of 30 seconds originally designed to track the fast variations in
atmospheric compositions during the triggered lightning flashes. In this study, we averaged these high-frequency observations
into a 3-min temporal resolution.

In addition to the ground-based observations, tropospheric ozone vertical profiles from satellite observations were also
analysed. The vertical distributions of ozone are measured by the AIRS instrument on the EOS Aqua satellite and the OMI
instrument on the EOS Aura satellite under the NASA Tropospheric Ozone and Precursors from Earth System Sounding





(TROPESS) project (Verstraeten et al., 2013; Fu et al., 2018; https://tes.jpl.nasa.gov/tropess/products/o3/, last access: 21

January 2022). The ozone profiles are produced via an optimal estimation algorithm using multi-spectra, multi-species, multi-sensors. These satellite-based ozone profiles have a spatial resolution of 13 km x 24 km with 26 vertical levels from the surface to 0.1 hPa, and the temporal resolution is 1 day. Fu et al. (2018) compared the joint AIRS+OMI against ozonesonde measurements, and they showed that the mean and standard deviation of their differences are within the estimated measurement

error of these space sensors (2-5 ppbv).

## 2.2 Meteorological observations and atmospheric reanalysis data

The operational radiosonde data from cites Jinan, Qingdao, and Weihai (Fig.1), along with ground-based automatic weather station observations, were utilized to capture the meteorological evolution responsible for the stratospheric intrusion. Regional radar mosaic products were produced and analysed using 3 Doppler radars including two S-band radars located in

Jinan and Qingdao and one C-band radar in Binzhou, because radar reflectivity and radial velocity are indicative of storm microphysical and dynamical structure, as well as the horizontal coverage and vertical extension of convection. Cloud-to-ground (CG) lighting flashes were also referenced to infer the storm development and intensity provided by a nationwide lightning detection network operated by the State Grid Electric Power Research Institute (Chen et al., 2012).

Three dimensional atmospheric MERRA-2 (The Modern-Era Retrospective Analysis for Research and Applications,

Version 2) reanalysis data were used to reveal the synoptic-scale evolutions impacted by typhoon In-fa (https://gmao.gsfc.nasa.gov/reanalysis/MERRA-2/, last access: 21 January 2022). MERRA-2 reanalysis has a horizontal resolution of $0.5° \times 0.625°$, 72 vertical levels from the surface to 0.01 hPa and a 3-hour update temporal cycle. The following gridded meteorological variables were extracted from MERRA-2. Dynamical variables including horizontal wind and vertical wind velocity were analysed to reveal dominant flow patterns when typhoon In-fa began dissipate. Potential vorticity (PV)

and relative humidity (RH), which are indicative of stratospheric intrusion, were used to track the variations in the tropopause height and the penetration of the stratospheric dryness.





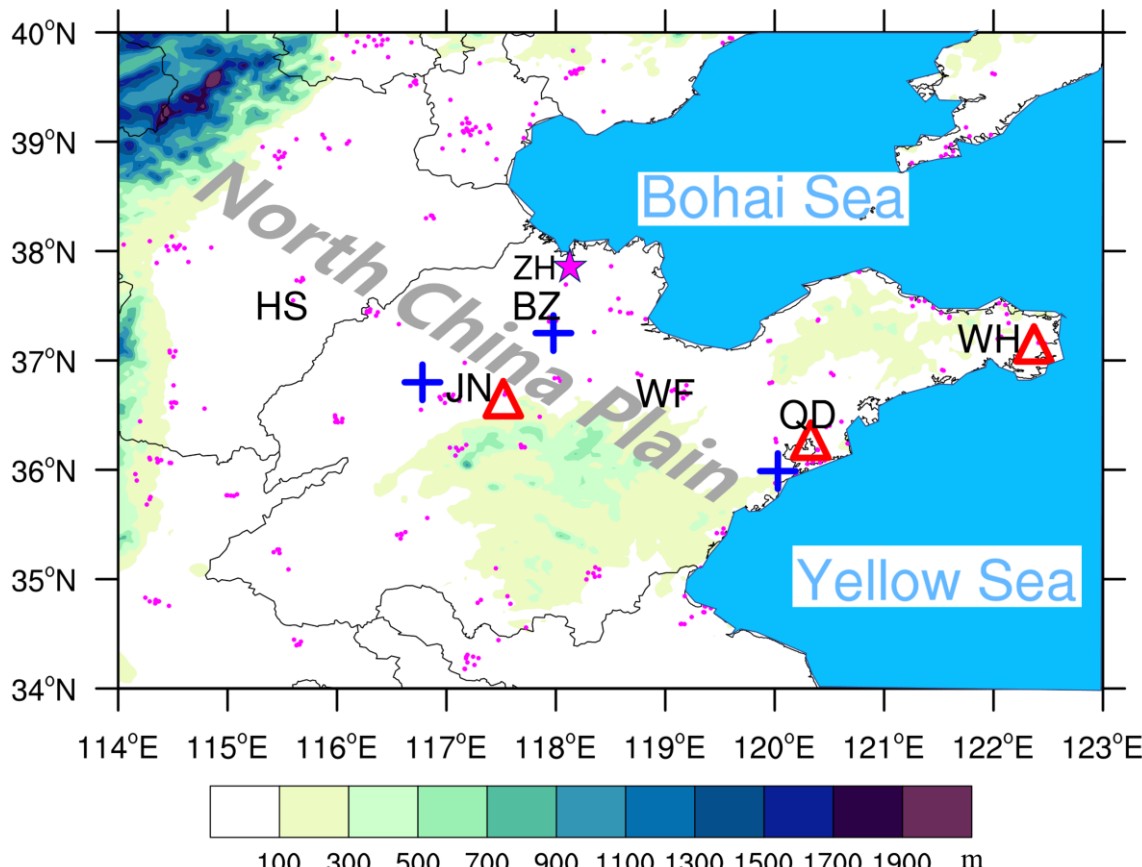

**Figure 1: Topography of North China Plain (NCP; color, unit: m) and locations of cities Hengshui (HS), Jinan (JN), Binzhou (BZ), Weifang (WF), Qingdao (QD) and Weihai (WH). Three radar stations, distributed in Jinan, Binzhou,**

**and Qingdao, are marked by blue cross symbols, and three radiosonde stations distributed in Jinan, Qingdao and Weihai are marked by red triangle symbols. The ground-based air quality monitoring stations are shown by magenta dots, and the station with high frequency measurement of air quality located in Zhanhua (ZH) county of Binzhou city is marked by a magenta star symbol. The locations of Bohai Sea and Yellow Sea are also indicated. The thin grey lines indicate the borders of provinces.**

**2.3 WRF simulations and FLEXPART backward trajectories**

The relatively coarse spatiotemporal resolution of observations and reanalysis data mentioned above cannot explicitly capture atmospheric processes at the storm scale, especially for the evolving convective dynamics responsible for the downward transport of ozone-rich airmass. For example, the 3-hour cycle of MERRA-2 reanalysis data can easily miss the details of MCS evolution and is insufficient for conducting storm-scale backward trajectory analysis. Therefore,

the dynamical evolution of the MCS under the influence of typhoon In-fa was simulated using the Weather Research



and Forecasting with the Advanced Research core (WRF-ARW, Version 3.9.1; Skamarock et al., 2008). The numerical simulation employed a two-way, three-domain nested grid cells. The outermost domain has 232×182 grids with a 27-km horizontal grid spacing and covers approximately the East Asia and neighbouring oceans. The inner domain has 490×430 grids with a 9-km horizontal resolution covering the entire China. The innermost domain is placed over the

NCP with 610× 610 grids and a 3-km horizontal resolution that guarantee to resolve the storm-scale features (Fig. S1). To explicit resolve the dynamical structure in the vertical direction, the number of terrain-following levels was set to 95, and the model top was set to ~50 hPa. As a result, the vertical spacing between each layer is approximately 100 m in the ABL (<1.5 km) and 200 m in the free atmosphere (between 1.5 km and 20 km).

In the simulation, the applied physics options in the WRF model include the Kain-Fritsch cumulus parameterization

scheme (Kain and Frisch, 1993), which was applied only to the outermost domain and inner domain but turned off for the innermost domain. The microphysical parameterization is the Morrison 2-moment scheme (Morrison et al., 2009), the boundary layer physics parameterization is the YSU scheme (Hong et al. 2006), and the land surface model is the Noah land surface model (Chen and Dudhia, 2001). For the longwave and shortwave radiation processes, the RRTM scheme (Mlawer et al. 1997) and the Dudhia scheme (Dudhia, 1989) were utilized. A 24-hour-period simulation starting from 08:00 LST (Local

Standard time, =UTC+8h) on 31 July covering entire lifespan of MCS was performed, which was initialized by the 0.5° and 3-hourly Global Forecast System (GFS) analysis of the National Centers for Environmental Prediction (NCEP). Simulation results of the innermost domain with a 3-km horizontal resolution were output every 3 min to analyse the evolution of storm-scale features.

Backward trajectories for the analysis of the surface ozone surge were simulated using the Flexible Lagrangian particle

dispersion model (FLEXPART) that work with the WRF model (FLEXPART-WRF, Version 3.3.2; Brioude et al., 2013; https://www.flexpart.eu/wiki/FpLimitedareaWrf, last access: 21 January 2022). The FLEXPART model (Stohl et al., 2005) was originally developed at the Norwegian Institute for Air Research in the Department of Atmospheric and Climate Research, and was further tailored to WRF models so that the model can be widely used to study the influence of meso-scale processes on pollution transport (e.g., Aliaga et al., 2021; Nathan et al., 2021). Based on the WRF simulation results of the innermost

domain, we conducted backward trajectory calculations using FLEXPART-WRF. Ten thousand particles were released at each defined location and timing, which would be described in the following section. The FLEXPART-WRF output was saved every 10 minutes to track the three-dimensional particle backward trajectories.

## 3 Confirmation of surface ozone surge with stratospheric origins

Before analysing this surface ozone surge case with stratospheric origins, it is beneficial to provide some statistics

of surface ozone background concentrations, especially over the NCP. Several researchers have pointed out that the summertime surface ozone over China shows a rapid increase since 2012 (Lu et al., 2018; Silver et al., 2018; Li et al.,




2019; Han et al. 2020), with estimated rates of ∼1.4–2.8 ppbv per year (Han et al. 2020; Li et al., 2020). In the 2021 summer, the daily mean and maximum 8 h average (MDA8) ozone concentrations in the NCP are 43.9 and 70.8 ppbv, respectively, while the mean nighttime ozone concentration is 36.6 ppbv. Fig. 2 shows a 10-day averaged surface ozone

concentration (from 27 July to 5 August 2021) in each city, used as the baseline for assessing ozone variations. Generally, the 10-day averaged ozone concentration in each city is close to the summertime mean ozone concentration of 45~50 ppbv. During 28-30 July 2021, under the cloudy conditions produced by typhoon In-fa, surface ozone was apparently lower than the 10-day average. After 31 July, as typhoon In-fa had moved out of the NCP and entered the Bohai Bay, the photochemical reactions accelerated as seen in the steady increase in surface ozone at daytime and normal diurnal cycles

since. However, instead of continuously decreasing after sunset, the concentrations of surface ozone over some cites in the NCP increased abruptly and intensively between 23:00 LST on 31 July and 06:00 LST on 01 August (between the vertical black lines in Fig. 2 and the zoomed-in Fig. S2), which were 40-50 ppbv larger than their corresponding monthly mean values at night and almost comparable to the daytime high ozone concentrations (Fig. S3). It is a common practice to use 25th percentile of ozone concentration distributions as a background value (e.g., Parrington et al., 2013), which yields

an even severer ozone enhancement in the surface. In cities Hengshui, Binzhou, Jinan and Weifang, a peak ozone concentration at nighttime reaching 80-90 ppbv appeared in succession, which was in accordance with the southeastward propagation of MCS (see section 4.2, where impacts of MCS on surface ozone will be addressed in details). While in the eastern cities such as Qingdao and Weihai (Fig. 2e-f) where convective activities were mostly absent, the ozone evolution at midnight was different from the cities experiencing storm passage shown in Fig. 2a-d.

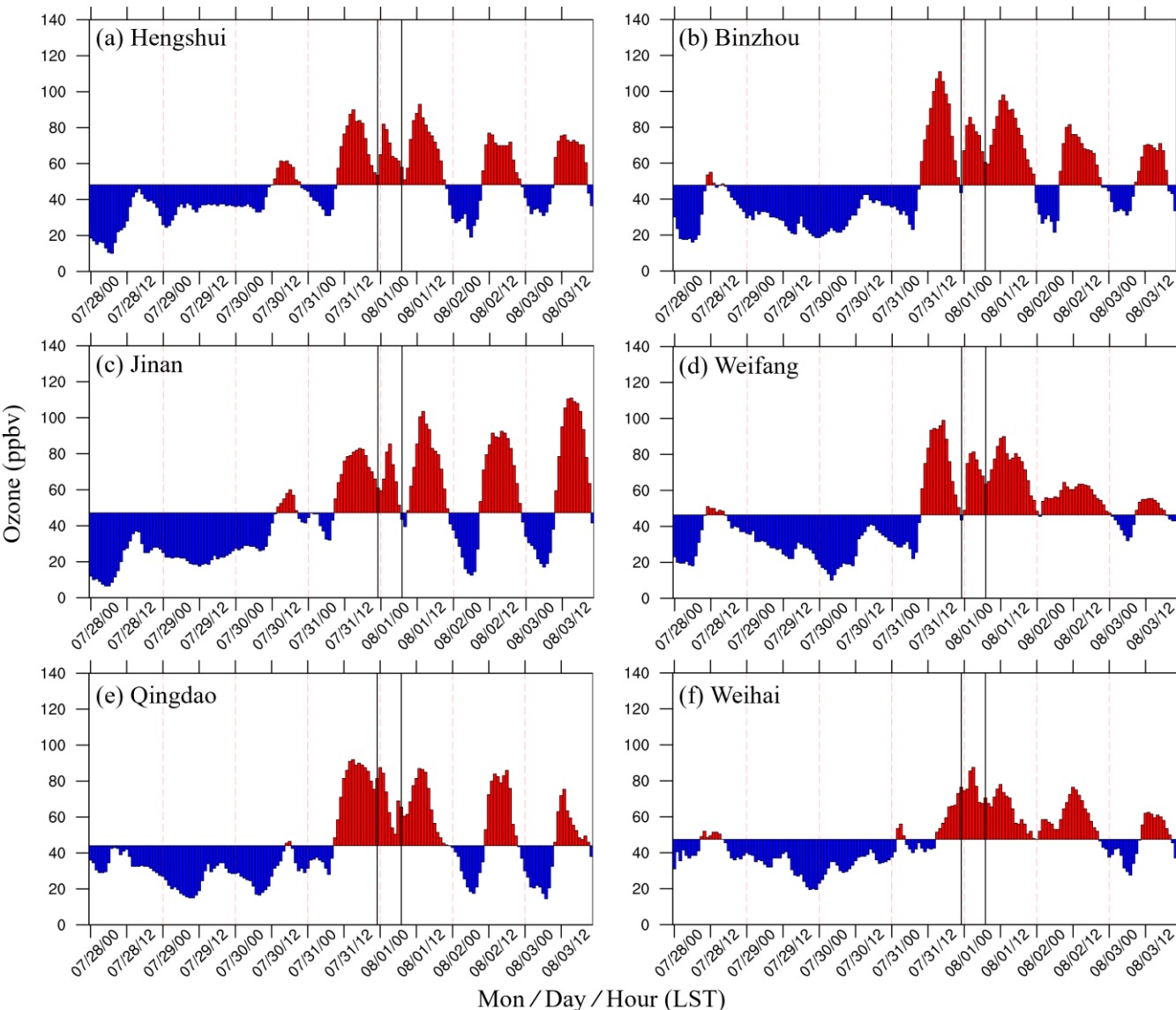

**Figure 2: Temporal variation in surface ozone concentrations (unit: ppbv) in local standard time (LST) from 28 July to 03 August 2021 using the 10-day averaged ozone value as a baseline for comparison in cities Hengshui, Binzhou, Jinan, Weifang, Qingdao, and Weihai. Positive (negative) departure from the 10-day averaged ozone concentration is shown in red (blue) color. The two vertical black lines represent the observed ozone surge period between 23:00 LST on 31 July and 06:00 LST on 01 August 2021. Daily cycles (0:00-0:00 LST) are denoted by vertical red lines.**

During the ozone surge period, an obvious decrease in surface CO was also observed. Fig. 3 shows the variations in surface CO with a 10-day mean concentration serving as the baseline. Though the temporal variations in surface CO were complex, a systematic low-concentration phase of CO appeared at midnight on 31 July (between the vertical black lines in Fig.





3 and the zoomed-in Fig. S4) when surface ozone surged (Fig. 2) and MCS took place. The surface CO concentrations were

greatly reduced in cities Hengshui, Binzhou, and Jinan, while the concentrations were reduced in Weifang during partial night time and were not reduced in Qingdao and Weihai. CO is often used as a tracer for both anthropogenic pollution and biomass burning (e.g., Pochanar et al., 2003; Lin et al., 2018); therefore, the high surface ozone synchronized with low CO in the time series confirms that the surface ozone surge was caused by stratospheric intrusions of ozone-rich and CO-poor airmass. The area impacted by stratospheric intrusions was larger than these cities covered, and was at least 500 km × 500 km based on the

nationwide atmospheric composition measurements.

Figure 3: Same as Fig. 2, but for surface carbon monoxide (CO) concentrations (unit: ppbv) from 28 July to 03 August 2021.



The observational results from the national monitoring network well revealed the surface ozone surge with
stratospheric origins spatially, though the ozone concentrations were averagely smoothed during each hour. To better
identify the magnitude and timing of surface ozone surge, high-frequency atmospheric composition measurements
collected during SHATLE field campaign at Zhanhua were analysed. Fig. 4 shows the 3-min variations in surface ozone
and CO concentrations relative to their 10-day averaged baseline concentrations. As a rural county of Binzhou city, the
ozone baseline concentration (approximately 60 ppbv) in Zhanhua was slightly higher than that of Binzhou city
(approximately 45 ppbv), while the CO baseline concentration, which is closely related to anthropogenic emissions, was
lower than Binzhou. The active photochemical reactions in the afternoon elevated ozone concentrations that fluctuated
between 100-120 ppbv. After sunset at 19:00 LST, surface ozone concentrations continuously fell down via titration
effect and dry deposition of vegetation, and thus was lower than its background concentration at 21:00 LST. However,
at 22:36 LST, the continuously decrease in surface ozone stopped. Instead, ozone concentrations surged abruptly from
31 ppbv to 80 ppbv in next 10 minutes, and maintained high in the next eight hours. The averaged surface ozone
concentrations in the night were 79 ppbv, and the maximum concentrations reached 93 ppbv at 01:54 LST on 01 August
2021. Based on the observations with finer temporal resolution, a synchronous reduction of surface CO concentrations
occurred exactly when ozone rose abruptly, which further confirmed that the ozone surge was caused by intrusions of
stratospheric airmass. The Chinese National Ambient Air Quality Standard for ozone exceedance level is 82 ppbv (Li et
al., 2020), and compared with the normal nighttime ozone concentrations (an average of 36.6 ppbv), the magnitudes of
surface ozone surge due to stratospheric intrusions were approximately 40-50 ppbv, which can pose great threats to
human health and agricultural crops and other plants.

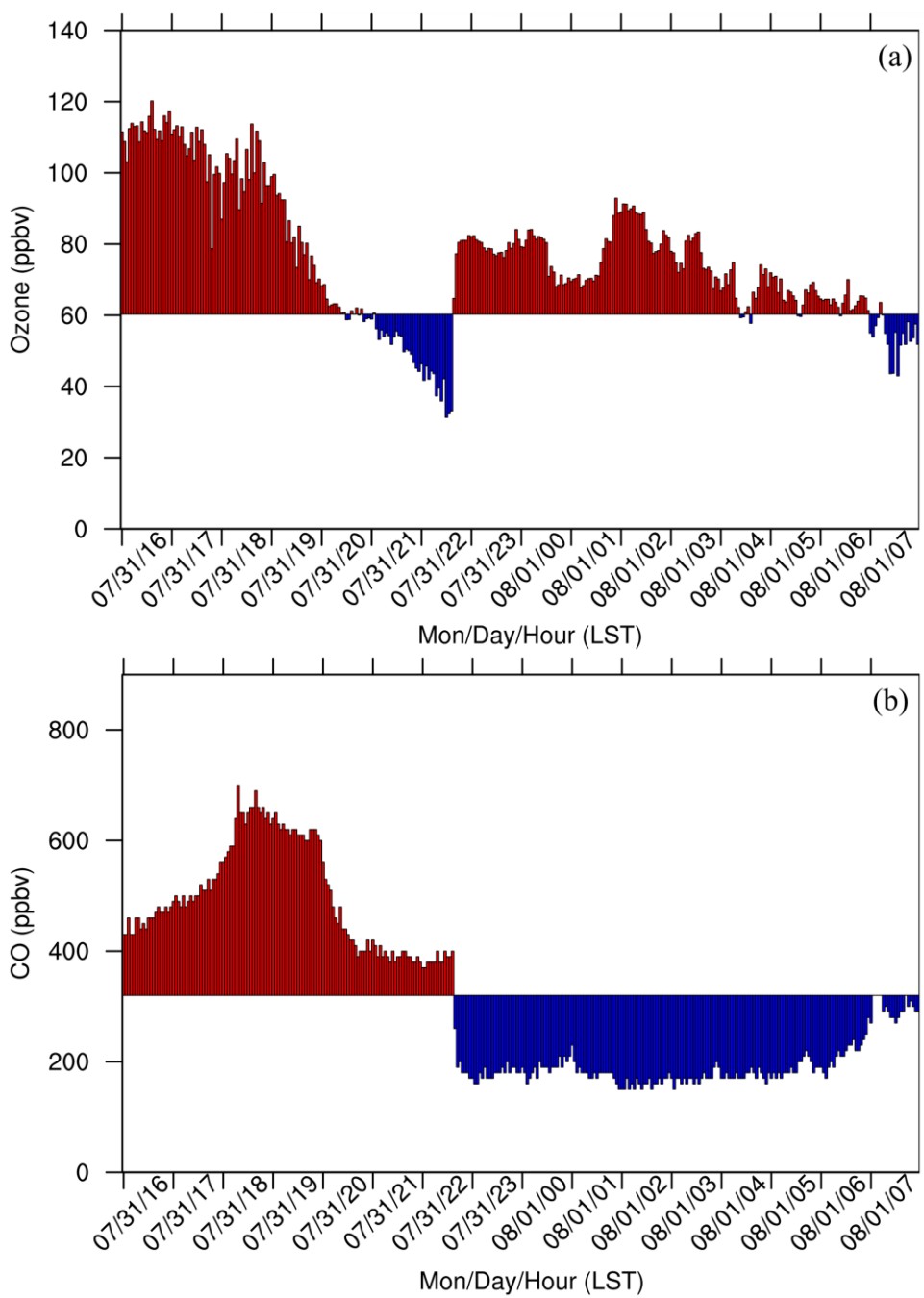

**Figure 4: Surface ozone and CO concentrations (unit: ppbv) at SHATLE field campaign site located in Zhanhua county**
**of Binzhou city, measured with a 3-min temporal resolution from 16:00 LST on 31 July to 08:00 LST on 01 August**
**2021. The 10-day averaged ozone and CO concentrations at the site are used as the baseline, and positive (negative)**
**departure from the 10-day averaged concentration is shown in red (blue) color.**





## 4 Multi-scale interactions responsible for the stratospheric intrusion

Several mechanisms have been proposed to explain higher tropospheric ozone concentrations than the normal. For
example, the STE associated with synoptic-scale dynamical exchange processes, such as tropopause folding near the polar jet
and subtropical jet (Stohl et al., 2003; Pan et al., 2004; Li et al., 2015a), cut-off low (Wirth et al., 1995; Li et al., 2015b), and
typhoon (Baray et al., 1999; Jiang et al., 2015; Preston et al., 2019; Chen et al., 2021), have been well studied. Local
photochemical production of ozone through emissions from biomass burning (Chan et al., 2003; Brioude et al. 2007) and
lightning-generated nitrogen oxides ($LNO_x$) (Cooper et al., 2006; Schumann and Huntrieser, 2007) are also able to increase
tropospheric ozone burden. In particular to this study, convection with overshooting tops can force subsidence air motions near
cloud edge due to mass continuity and hence transport stratospheric ozone-rich air downward (Hu et al., 2010; Pan 2014;
Phoenix et al. 2020). In this surface ozone surge event associated with stratospheric intrusions, the dominant atmospheric
processes are the dying typhoon In-fa and the local MCS, with no influences from ozone precursors from biomass burning or
$LNO_x$. In the following, we provide detailed analyses on the interactions between synoptic-scale and convective-scale
processes that finally bring ozone originated from the stratosphere to the surface and lead to the intensive midnight ozone
surge.

### 4.1 Large-scale descent of stratospheric airmass attributed to dying typhoon In-fa

Previous studies indicate that typhoon can perturbate the tropopause and hence induce stratospheric intrusion that
bring ozone-rich airmass to the lower troposphere and even ABL. Using a large ensemble of landfalling typhoon cases,
Chen et al. (2021) found significant positive ozone anomalies at the middle and upper troposphere due to stratospheric
intrusion when typhoons are intensive, and negative ozone anomalies within the entire troposphere when typhoons have
made landfall. While in this case study, typhoon In-fa shows different features from the ensemble-averaged behaviours.
The In-fa made landfall in southern China approximately at 12:00 LST on 25 July 2021 with a maximum wind speed of
38 m s$^{-1}$ (typhoon category) and gradually weakened along its northward passage over land. At 08:00 LST on 29 July
2021, In-fa entered the NCP (magenta cross symbols in Fig. 5a and Fig. S5) with a maximum wind speed of 15 m s$^{-1}$
(TD category) and propagated slightly northeastward to the Bohai Bay (Fig. 1). The monitoring of In-fa's track and
intensity by the Meteorology Administration of China was terminated after 20:00 LST on 30 July 2021, given its weaker
intensity than TD category. Consequently, In-fa maintained its existence over land for more than 5 days (128 hours). Fig.
5a shows the 700-hPa vertical air motions superimposed on the 850-hPa horizontal wind flows at 20:00 LST on 30 July
2021 based on MERRA-2 reanalysis data. Though the intensity of In-fa declined steadily and could not even satisfy the
TD category, In-fa was still capable of maintaining systematic upward air motions with anticlockwise circulations at the
Bohai Bay and inducing downward air motions over land. In the vertical direction (Fig. 5b), the downward air motions





over land were deep extending from surface to 500 hPa. The dynamical tropopause represented by the 2.5-PVU (potential

vorticity unit, 1 PVU $= 10^{-6}$ K m$^2$ s$^{-1}$ kg$^{-1}$) contour line mainly located at approximately 100 hPa, and the stratospheric

dryness with relative humidity (RH) less than 30 % had reached around 300 hPa. On the next day, a significant downward

placement of 2.5-PVU dynamical tropopause and dryness occurred under the influences of In-fa (Fig. 5c-d). At 14:00

LST on 31 July 2021, the tropopause descended to 300 hPa and the dry airmass filled the upper troposphere above 500

hPa, yielding great potential for stratospheric intrusions even though the In-fa was in dissipation stage.

**Figure 5: (a) Vertical velocity (shaded; 0.01 Pa s$^{-1}$) at 700 hPa overlaid with 850-hPa horizontal wind flows (grey vector; reference vector is 12 m s$^{-1}$) at 20:00 LST on 30 July 2021. The magenta cross symbols represent the last six track information of typhoon In-fa. (b-d) Cross sections of vertical velocity (shaded; 0.01 Pa s$^{-1}$), relative humidity (black solid lines with values of 10% and 30 %) and the 2.5-PVU dynamical tropopause height (magenta solid lines) at 20:00**



**LST on 30 July (b), 08:00 LST on 31 July (c), and 14:00 LST on 31 July 2021(d). The cross sections are performed**
**along the black dashed line in Fig. 5a.**

Vertical profile observations can reveal details of the large-scale descent of stratospheric airmass attributed to the dying typhoon In-fa. Water vapor and ozone are tracers commonly used to detect stratospheric airmass. Previous observations collected at mountain peaks suggested that the frequency of stratospheric intrusions is at minimum in summer, and stratospheric intrusions that directly influence ozone concentrations below 700 hPa are rare (Elbern et al. 1997; Stohl et al.
2000). Therefore, we averaged the moisture and ozone of the airmass below 700 hPa over the 10 days (28 July to 3 August 2021) and used the averages as the baseline to track stratospheric intrusions induced by the dissipating In-fa. Fig. 6 shows the vertical profiles of dewpoint depressions (T-$T_d$) relative to the 10-day averaged baseline between the surface and 700 hPa using radiosonde observations collected in Jinan. Consistent with the continuous downward penetration of stratospheric dryness shown in Fig. 5, the dry airmass associated with large dewpoint depressions over Jinan sunk down
to 900 hPa at 20:00 LST on 31 July, which further replaced the low-level moist air and reached the ground level as seen in the profile at 08:00 LST on 01 August. The timing of surface ozone surge in Jinan was in agreement with variations in atmospheric moisture profile. Radiosonde observations at Qingdao and Weihai also confirmed the large-scale descending of stratospheric dryness impacted by In-fa, however, the near-surface airmass in Qingdao and Weihai were moister than their baseline values (Fig. S6 and S7) on 31 July and 01 August, suggesting weaker impacts of stratospheric
intrusion at the surface in these cities than Jinan.







**Figure 6: Profiles of dewpoint depressions (T-T$_d$, unit: °C) from Jinan radiosonde observations at (a) 20:00 LST on 29 July, (b) 20:00 LST on 30 July, (c) 08:00 LST, (d) 20:00 LST on 31 July, (e) 08:00 LST 01 and (f) 20:00 LST on 01 August. The 10-day averaged dewpoint depressions between the surface and 700 hPa are used as the baseline, and positive (negative) departure from the 10-day averaged value is shown in red (blue) color.**

Behaviours of vertical ozone profiles under the influence of In-fa were examined using satellite ozone observations. Fig. 7 shows the mean profiles of ozone concentrations over the NCP against the baseline ozone concentration (56 ppbv) averaged between the surface and 700 hPa based on TROPESS AIRS L2 ozone products. Compared with the ozone profile on 29 July, a significant increase in tropospheric ozone occurred in the following three days. Impacted by the stratospheric ozone-rich



airmass, the positive ozone anomalies relative to the baseline concentration extended downward to the lower troposphere. Despite possible bias of AIRS ozone profiles especially at low levels, the relative variations in vertical ozone concentrations between those days clearly revealed the large-scale downward propagation of ozone enhancement under the influence of dissipating In-fa. The concurrent trends of atmospheric moisture and ozone provide a piece of clear evidence that the stratospheric airmass had descended to the middle-to-low troposphere (at least 900-500 hPa) during the evening on 31 July

over the NCP, which was adequate to initiate the subsequent vigorous surface ozone surge.

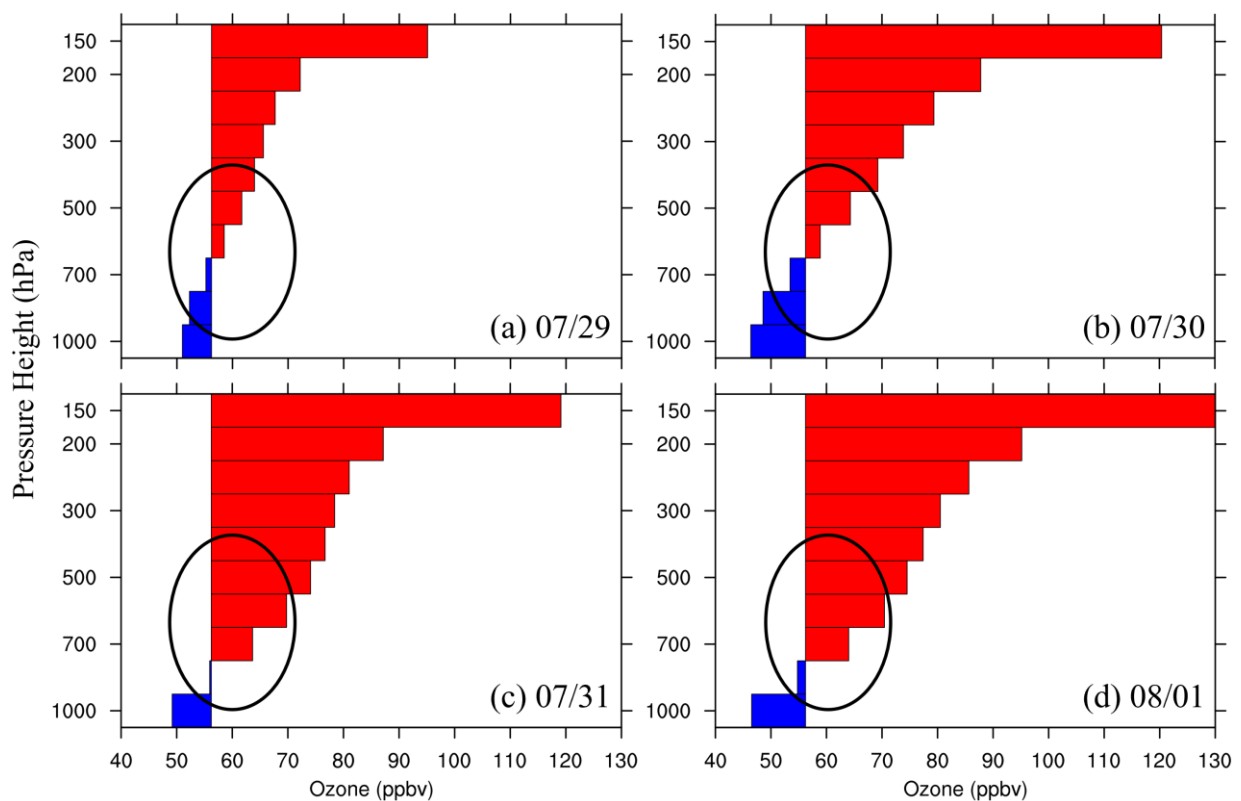

**Figure 7: Spatially averaged profiles of ozone concentrations (unit: ppbv) over the NCP from using the TROPESS AIRS L2 ozone products on (a) 29 July, (b) 30 July, (c) 31 July, and (d) 01 August. The 10-day averaged ozone concentrations between the surface and 700 hPa over the NCP are used as the baseline, and positive (negative)**

**departure from the 10-day averaged concentrations is shown in red (blue) color.**

## 4.2 Convection-facilitated stratospheric intrusion and transport pathways of ozone-rich airmass

The above analyses reveal a large-scale downward intrusion of stratospheric airmass to the lower troposphere under the influence of dissipating typhoon In-fa. However, the responses of surface ozone concentrations differed spatially (Fig. 2),



which leave an important question of how stratospheric ozone-rich air was transported to the surface. To be more exact, what

are the mechanisms responsible for the final descending of stratospheric airmass to the surface? Previous studies indicated that deep convection with overshooting tops can effectively transport stratospheric ozone-rich air to the surface (e.g., Poulida et al., 1996; Hu et al., 2010; Pan et al., 2014). Such convective redistribution of ozone in vertical profile is driven by dynamical processes, in which vigorous upward motions penetrate into the stratosphere and induce compensating subsidence of stratospheric ozone-rich air. A bow-echo MCS formed and passed through the NCP at night on 31 July. In this section, we will

illustrate how the MCS facilitated the final descending of stratospheric airmass to the surface.

Fig. 8 shows the hourly radar mosaic observations at the evening of July 31-August 1, 2021 during which ozone concentrations at the ground stations exceeded 80 ppbv. At 20:00 LST (1 hour after sunset; Fig. 8a), two convective cells were located at southwest and northeast of Hengshui, and quite a lot stations still maintained high ozone concentrations accumulated from the daytime photochemical reactions. The northeastern convective cell developed rapidly with increasing horizontal areal

coverage and evolved into bow-echo MCS, while the southwestern cell gradually weakened (Fig. 8b-f). The number of stations with high ozone concentrations decreased as a result of titration effect and dry deposition, however, significant surface ozone enhancement occurred in the convective downdraft regions along with the bow-echo MCS development and propagation. For example, at 00:00 and 01:00 LST on 01 August, surface ozone concentrations increased abruptly, coincident with a CO decrease, when the bow echoes passed through Binzhou and Jinan. As the bow-echo MCS kept travelling southeastward, the

downstream regions of convection such as Weifang experienced convective downdrafts and hence ozone surge subsequently (Fig. 2). While in regions where convective activities were weak or absent such as Qingdao and Weihai, despite the high ozone episode lasting more than several hours, the surface ozone enhancement at midnight were subtle suggesting that stratospheric airmass did not reach the surface.





**Figure 8: Observed radar reflectivity structure (shaded; dBZ) of the bow-echo MCS occurred at night on 31 July 2021. Stations with high ozone concentrations are mapped by large solid circles in different colors.**

Bow echoes are the bow-shaped segment of radar reflectivity structures within squall lines that can persist for several hours and produce damaging winds near the apex of the bow, particularly when the rear inflows descend to the surface. The rear inflows originate from the rear anvil cloud of the stratiform region and descend toward the leading convective line. They are driven by the diabatic cooling processes at the middle levels, in which precipitation particles falling from the stratiform clouds evaporate, melt and cool the air (Keene and Schumacher, 2013; French and Parker, 2014). With reference to radar radial





wind observations (not shown here), the descending rear inflows of bow echoes exceeded 25 m s$^{-1}$ from the trailing cloud region and hence brought down the stratospheric ozone-rich airmass that located at 900-500 hPa. Different from the case studies of deep convection with overshooting tops reaching stratosphere (e.g., Pan et al., 2014), the bow-echo MCS in this case

were relatively weak and did not penetrate to the tropopause altitudes. Fig. S8 shows the temporal evolution of vertical radar reflectivity profiles over Jinan and Binzhou. Following the standard World Meteorology Organization (WMO) lapse-rate criterion (Reichler et al., 2003), the thermal tropopause height was 15.8 km based on the nearest sounding collected in Jinan station at 20:00 LST on 31 July 2021. The overall radar reflectivity structure over Jinan and Binzhou did not reach the thermal tropopause height, and the strong radar reflectivities were confined below 6 km altitude (480 hPa, -9 °C) suggesting limited

vertical extension of convective storms. Lightning flashes are indicative of vertical development of thunderstorm. A total of 362 cloud-to-ground lightning flashes were detected from 21:00 LST 31 July to 06:00 LST 01 August 2021 within 50-km radius of Zhanhua station. It is inferred that the bow-echo MCS was weakly electrified due to shallow extension above the freezing level. Owing to the pre-existed stratospheric ozone-rich airmass located in the lower troposphere under the influences of the dying typhoon (Fig. 5), the middle level rear inflows can facilitate the downward transport of ozone to the surface even

though the convection was relatively shallow and weak. This case provides new insights into the interactions between synoptic-scale and meso-scale atmospheric processes that enable the direct stratospheric intrusion to the surface.

To better depict the convective-scale transport pathways facilitating the final descending of stratospheric ozone-rich airmass to the surface, high-resolution WRF simulations of the bow-echo MCS were performed and used to drive backward trajectories using FLEXPART model. Fig. 9 shows the WRF-simulated radar reflectivity structure of the bow-echo MCS. As

compared with the radar observations shown in Fig. 8, the WRF simulation reproduced the two convective cells distributed in the southwestern and northeastern regions of Hengshui, respectively (Fig. 9a). Despite slightly overforecasted convection distribution, the WRF simulation well captures the subsequent dissipation of the southwestern cell and the evolution of the northeastern cell into bow echoes passing through Binzhou and Jinan (Fig. 8b-f vs. Fig. 9b-f). Furthermore, the simulations were quantitatively evaluated against observations as represented in the categorical performance diagram, which is an

evaluation technique commonly used in convective-scale data assimilation and forecasting (Roebber, 2009). The performance diagram merges multiple metrics including bias, POD (probability of detection), SR [success ratio, = 1-FAR (false alarm rate)], and CSI (critical success index) into one graph, and simulations lie on the upper-right corner of the diagram. As shown in Fig. S9, the POD for 30-dBZ radar reflectivity threshold exceeds 0.7, and the SR and CSI increased steadily as the MCS pass through Binzhou and Jinan suggesting the satisfactory simulations from WRF. Given the general agreement between the

simulations and observations, we use the output from this high-resolution model to address the transport pathways of stratospheric ozone-rich airmass from the upper troposphere to the surface.

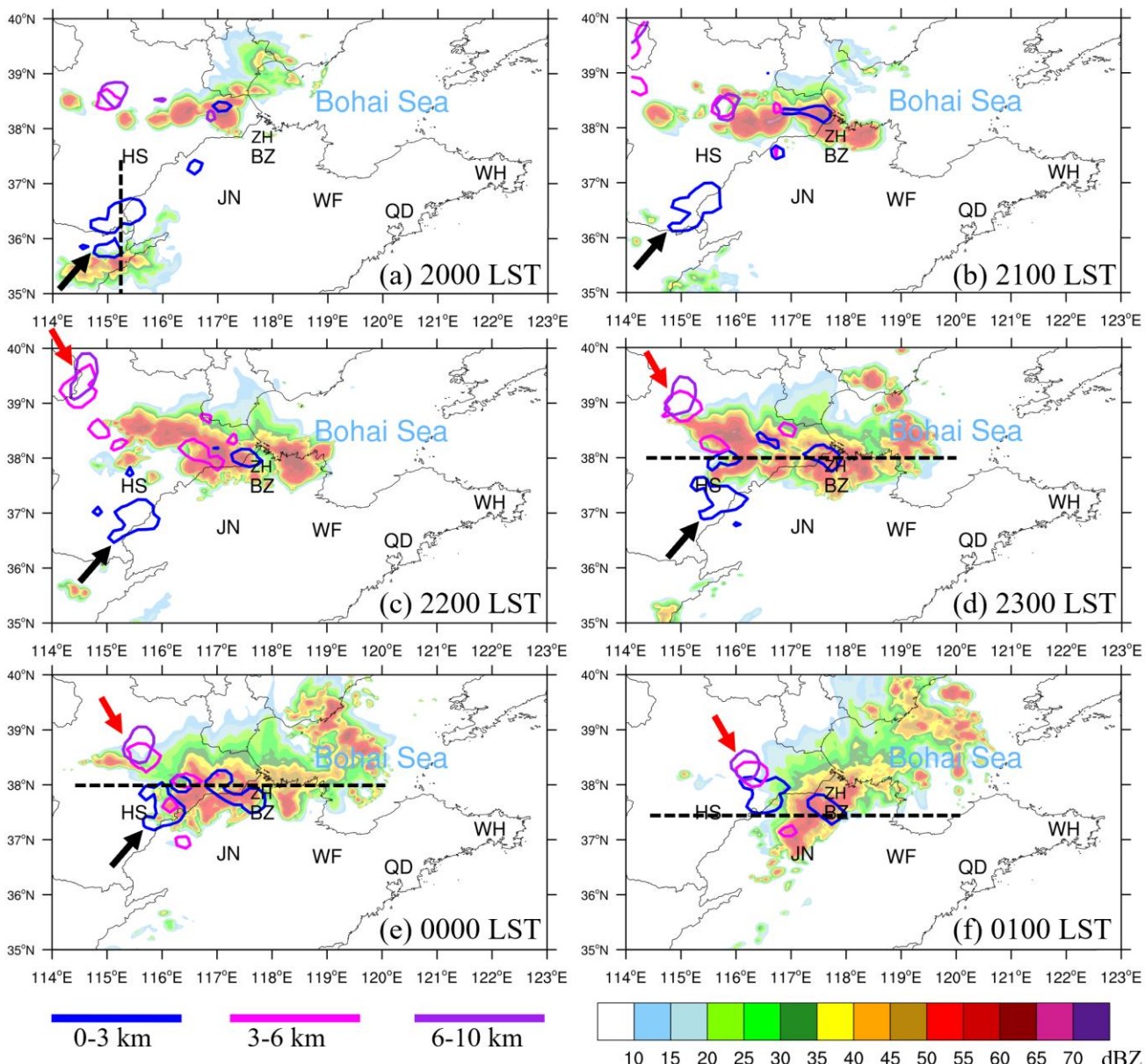

**Figure 9: WRF-simulated radar reflectivity structure (shaded; dBZ) of the bow-echo MCS occurred in the night of 31 July 2021. The solid lines represent regions with high tracer particle counts released in the Binzhou between 0-3 km (blue lines), 3-6 km (magenta lines) and 6-10 km (purple lines). The black dashed lines are the cross-section lines used in Fig. 11. The red and black arrows highlight the movements of the tracer particles.**

Remember that in this event of large-scale descending of stratospheric ozone-rich airmass, surface ozone concentrations increased abruptly and vigorously at cities Jinan, Binzhou and Weifang (Fig. 2), where bow-echo MCS passed through, while in cities such as Qingdao and Weihai where convection was weaker or absent, surface ozone enhancement or surface CO





reduction was weak (Figs. 2 and 3). Through the FLEXPART simulations driven by meteorology field from WRF model, two scenarios concerning ozone transport were designed and the backward trajectories of tracer particles were analysed. In the first scenario, tracer particles were released in Binzhou between 1000-950 hPa at 04:00 LST on 01 August when the stratospheric airmass had reached the surface. In the second scenario, tracers were released at Qingdao in order to examine the contribution from the convection to the surface ozone surge. Fig. 10 shows the temporal variations in vertical tracer particle counts in each

scenario with reference to the three-dimensional location of tracers in backward time. In the Binzhou release scenario, the upper boundary of vertical distributions of tracers was approximately 11 km on 31 July 2021, while the thermal tropopause height was 15.8 km. Therefore, it can be inferred that the stratospheric ozone-rich airmass that reached the surface were not freshly produced from wrapping and shedding of stratospheric airmass by the MCS (Pan et al., 2014). Before convection formed (09:00-20:00 LST here), the tracer particles concentrated between 4-6 km, corresponding to the large-scale intrusion

of stratospheric airmass toward the middle-to-low troposphere under the influence of dying In-fa. During this slowing descending phase, the distribution of tracers was typical of filamentary structure due to the weak large-scale descending motions. As convection developed and evolved into bow-echo MCS, there were two periods with rapidly descending tracers. The former rapidly descending phase took place at 20:00 LST at 2 km, while the later occurred at 23:00 LST at 3 km, which would be analysed in details in the following part. In the Qingdao release scenario, though the distribution of tracers extended

upper to 10 km, quite a large portion of tracers remained below 1 km on 31 July 2021, suggesting that surface ozone in Qingdao mainly originated within the boundary layer and hence were faintly influenced by the stratospheric airmass. The distinctly different distribution patterns between the two scenarios indicated that convection had a considerable influence on facilitating the final descending of stratospheric airmass to the surface.





**Figure 10: Temporal variations in vertical tracer particle counts released at (a) Binzhou and (b) Qingdao. Tracer particle counts with a value of 100 (equalling 1 % of the total number of released tracers) are highlighted by magenta lines. The black arrows highlight the two rapidly descending phases of stratospheric airmass.**





A key scientific aspect concerning stratospheric impacts on surface ozone is how the stratospheric airmass reaches the surface. The backward trajectories of tracers during the two rapidly descending phases in the Binzhou release scenario were used here to address the convective-scale transport pathways of stratospheric airmass to the surface. We separated the distributions of tracer particles into the low (0-3 km), middle (3-6 km) and high (6-10 km) levels, and superimposed them on the radar reflectivity evolution of the MCS (Fig. 9). During the first rapidly descending phase at 20:00 LST on July 31, a low-level region with high tracer particle counts (black arrow in Fig. 9) appeared in the northern flank of the southwestern convective cell and propagated northeastward to Binzhou. The cross-section of southwestern cell and tracer distributions (Fig. 11a) indicate that the low-level tracers were transported by the widespread outflow winds between 0-3 km. At 20:00 LST, the stratospheric ozone-rich airmass had likely been transported to the surface by the dissipating of the southwestern cell (referred to the high surface ozone concentrations in Fig. 8a-c), and the ozone-rich airmass was transported horizontally by the downdraft outflows of the southwestern cell toward Binzhou.

In addition to the horizontal transport of ozone at low-levels by the southwestern cell, the middle- and high-level regions with high tracer particle counts expanded in the rear part of the northeastern convective cell that evolved into bow echoes. During the second rapidly descending phase, a significant rearward-sloping configuration of regions with high tracer particle counts was noticeable from low to high levels (red arrow in Fig. 9). We performed cross-section analyses of the bow-echo MCS (Fig. 11b-c), and the results clearly show a rearward pathway through which the stratospheric ozone-rich airmass was transported to the surface by the rear inflows descending from stratiform clouds to the leading convective line. Though the tropopause was perturbated and hence deformed by convective dynamics, the bow-echo MCS did not penetrate the tropopause significantly and were not likely to bring down fresh stratospheric air from the cloud edges. Instead, because of the pre-existed stratospheric airmass located in 3-6 km, rear inflows of the MCS originating from the middle level could easily facilitate the downward transport pathways for stratospheric ozone. Previous studies documented that the downward transport of stratospheric ozone can occur both in the rearward anvil and forward anvil (Stenchikov et al., 1996; Pan et al. 2014), and the transport in forward anvil is more rapid (Phoenix et al., 2020). While in this case, there only existed rearward transport pathways for stratospheric ozone-rich airmass, which was probably due to the relative weak and shallow structure of MCS.





**Figure 11: Cross sections of WRF-simulated radar reflectivity structure (shaded; dBZ), tracer particle distributions from FLEXPART model (magenta lines) and wind flows (vectors) at (a) 20:00 LST, (b)23:00 LST on 31 July, (c) 00:00 LST and 01:00 LST on 01 August 2021. The red solid circles represent the thermal tropopause height calculated from WRF simulations. The cross-section lines are shown in Fig. 9.**

## 5 Conclusions and discussions

In this paper, we focused on analysing an unusual surface ozone surge event with stratospheric origins occurring at night over the North China Plain (NCP), where both population and agricultural vegetation are high while the impacts of stratospheric intrusions on surface ozone are relatively less studied. Based on ground-based atmospheric composition





observations, satellite ozone profile products, meteorological data including radiosonde and radar observations and MERRA-2 reanalysis products, we confirmed the stratospheric influences of this unusual nighttime surface enhancement and documented the evolution and magnitude of the surface ozone surge in detail. The mechanisms responsible for this direct stratospheric intrusion reaching the surface and the transport pathways of ozone-rich airmass were investigated using high-resolution model simulations and backward trajectory analyses. The conclusions are drawn as follows:


(1) Evolution and magnitude of the surface ozone surge. The surface ozone surge mainly occurred between 23:00 LST on 31 July and 06:00 LST on 01 August 2021 over the NCP and swept southeastward through a large spatial coverage (~ 500 km × 500 km). Instead of decreasing continuously after sunset as normal, surface ozone increased abruptly and significantly. Surface ozone concentrations at midnight in cities such as Hengshui, Binzhou, Jinan and Weifang reached 80-90 ppbv in succession that were nearly twice as large compared to the baseline ozone concentrations. Referring to the high-frequency measurements, the ozone concentrations at Zhanhua station surged from 31 ppbv to 80 ppbv within 10 minutes, indicating that the stratospheric airmass can enhance surface ozone by 40-50 ppbv within a short time period. A concurrent vigorous decline of surface CO concentrations was observed, which confirmed that the surface ozone surge was caused by stratospheric intrusion of ozone-rich and CO-poor airmass. This is further confirmed by the vertical evolutions of humidity and ozone profiles at night, based on radiosonde and satellite data, respectively. In terms of magnitude, covering areas, abruptness, and duration, such a stratospheric impact on surface ozone is rarely documented.



(2) Mechanisms for the direct stratospheric intrusion to reach the surface. The vigorous surface ozone enhancement was induced by the multi-scale interactions between the dying typhoon In-fa and local MCS. Though the typhoon was in dissipation stage after a 5-day travel over land, it could still perturbate the tropopause and maintain the downward motions over the NCP that brought down dry and ozone-rich airmass as seen in the reanalysis data as well as moisture and ozone profiles. Before the local MCS occurred, the airmass with stratospheric origins had descended to the middle-to-low troposphere (900-500 hPa) over the NCP. The local bow-echo MCS facilitated the final descending of stratospheric airmass to the surface through the development of convective downdrafts. Significant surface ozone enhancement occurred in the convective downdraft regions during the development and propagation of the bow-echo MCS. While at stations where convective activities were weak or absent, the surface ozone and CO evolutions at the midnight were not in a high-ozone and low-CO pattern, suggesting that stratospheric airmass did not reach the surface.



(3) Transport pathways of ozone-rich airmass to the surface. In the face of pre-existed stratospheric airmass, the rear inflows of bow-echo MCS transported the ozone-rich airmass downward from the mid-level rear stratiform cloud to the leading convective line and eventually to the surface. Compared with the large-scale descending processes associated with the dying typhoon, the convection-facilitated transport processes of ozone were rapider. Based on high-resolution simulations and trajectory analysis, two convective-scale transport pathways responsible for ozone enhancement at surface stations were identified. The direct pathway was the vertical transport of ozone through rear inflows of convection, which can effectively bring down the ozone-rich airmass to the surface. The indirect pathway




mainly involved the horizontal transport of ozone by mature storms that had already brought down the ozone-rich airmass.

Previous studies found the association of stratospheric intrusions with strong convection, for example, intensive typhoons before making landfall and thunderstorms with over-shooting tops. This case study provides new insight into the interactions between synoptic-scale and meso-scale atmospheric processes that enable a direct stratospheric intrusion to reach the surface.

Though the typhoon in this case was in dissipation stage, and the local MCS were relatively shallow and weak without obvious over-shooting feature, step by step, the typhoon induced stratospheric intrusion reached the middle-to-low troposphere, and then MCS facilitated the intrusion to reach the surface. This kind of multi-scale stratospheric intrusions can pose unexpected threatens of large ozone enhancement to human health and vegetation growth. Over a short timescale, timely warning and prediction of such ozone surges associated with multi-scale interactions of atmospheric processes are important for ecosystem

wellbeing, which require a deeper understanding in the mechanisms for convective redistribution of vertical ozone profiles in the atmosphere. In addition, the chemical consequences of vigorous ozone surges on air quality should be further explored in order to issue appropriate management policies. Over longer timescales, a proper analysis on the frequency and magnitude of convection-driven (including weak convection) ozone changes is crucial to better differentiate the natural and anthropogenic contributions to the rapid ozone increase in the region (Lu et al., 2018; Li et al., 2019; Han et al. 2020). Since such dynamical

transports of ozone associated with convection are inexplicitly expressed in global chemistry climate models, the stratospheric ozone input to the troposphere and ABL is probably underestimated (Pan et al., 2014). In the context of global warming, the frequency and intensity of convection is projected to increase, which underscores the necessity of considering these processes into the global model of atmospheric chemistry.

**Data Availability Statement**

The surface air pollutant observations obtained from the China National Environmental Monitoring Centre can be obtained from http://106.37.208.233:20035/. The MERRA-2 reanalysis meteorological data can be downloaded from https://gmao.gsfc.nasa.gov/reanalysis/MERRA-2. Satellite-based ozone vertical profiles measured by the AIRS and the OMI under the NASA TRopospheric Ozone and Precursors from Earth System Sounding (TROPESS) project are obtained from https://tes.jpl.nasa.gov/tropess/products/o3/. The applied Weather Research and Forecasting with the Advanced Research core

(WRF-ARW, Version 3.9.1) model is open-source code in the public domain maintained by the National Center for Atmospheric Research (NCAR; https://www2.mmm.ucar.edu/wrf/users/download/get_source.html). The Flexible Lagrangian particle dispersion model (FLEXPART) that work with the WRF model (FLEXPART-WRF, Version 3.3.2) is downloaded from https://www.flexpart.eu/wiki/FpLimitedareaWrf. The data and model output are available for scientific investigations upon request.



## Author contributions

ZC and JL designed the study and performed the research with contributions from all co-authors. YS, XC, ZC and XL collected the observations and analysed the data. XQ runs the field campaign of Shandong Triggering Lightning Experiment (SHATLE), and contributed to the backward trajectory analysis of tracers. ZC and JL wrote and revised the paper, with input from XC and MY. All authors commented on drafts of the paper.

## Competing interests

The authors declare that they have no conflict of interest.

## Acknowledgments

This work was supported by the National Natural Science Foundation of China (Grants 42105079). The computing resources used in this study were provided by Fujian Normal University High Performance Computation Center (FNU-HPCC). Specifically, we thank Dr. Rubin Jiang and Zongxiang Li for maintaining the atmospheric composition instruments in the field campaign.

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
