# Peer review of "Transport of substantial stratospheric ozone to the surface by a dying typhoon and shallow convection"

_Atmospheric Chemistry and Physics, 2022_

## Referee Comment (RC3)

**General Comments**

Overall, this is an interesting study that adds insight into the role that typhoons and mesoscale convective systems play in transporting stratospheric ozone to the surface. The science of this study is sound, the paper is well organized, and the figures nicely complement the text. However, there is poor grammar throughout the paper which makes it difficult to read at times. I include specific comments below that address many of these grammatical errors. My comments below are mainly minor in scope. As a result, I fully support acceptance of this paper once the following comments are addressed. I think this paper will be a good addition to the ACP journal.

**General Comments**

You should add a table that lists basic parameters you used for your WRF simulation. This is standard practice when running a model simulation.

You should capitalize "Typhoon" since you are referring to a specific one by name. You should do this throughout the paper anytime you reference "Typhoon In-fa".

Be consistent with how you refer to water "vapor". You go back and forth between "vapor" and "vapour" throughout the paper. These are both correct so select one and stick with it.

Make sure to include articles like "a" and "the" before referring to a singular noun. I did my best to address some of these specific instances below but I did not discuss all of them below.

Spell out "Figure" if it begins a sentence.

**Specific Comments**

Line 14 – Change "…at night…" to "…during the night…"

Line 16 – Provide the time units (LT or UTC) for 23:00 and 6:00

Line 16 – Change "01 August, 2021" to "1 August 2021". There are other instances of this throughout the paper so fix date formats accordingly.

Line 19 – Change "…of…" to "…of a…"

Lines 20-21 – Reword this sentence. You make a good point, but this is lost here because of grammar issues.

Lines 23,66 – You should define the MCS acronym as MCSs because you refer to "systems" which is plural. Once this is defined then you can use "MCS" in the future when referring to just a single mesoscale convective system.

Line 26 – Change to "…still bring down a stratospheric…"

Line 28 – Change "pre-existed" to "pre-existing"

Line 29 – Change "descending" to "descent"

Line 39 – Change to "…contains only 5-10% of atmospheric ozone, as well as high water vapor and CO concentrations due…"

Line 43 – Be careful here. You refer to deep convection which is singular but then you use "they" later in the sentence. Be consistent with your grammar.

Line 48 – Change to "…in the UTLS."

Line 51 – Change to "…from the ABL to the UTLS…"

Line 53 – Change to "…for the radiation balance…"

Line 58-62 – This sentence is too long. I suggest breaking this up into two separate sentences.

Line 80 – Change "Compare" to "Compared" since the comparison occurred in the past tense.

Line 82 – Change to "Such a significant…"

Line 85 – Change "…in 2021 summer…" to "…in Summer 2021."

Line 90 – Replace "unneglectable" with a better word here

Line 91 – Use a singular verb "was" instead of "were" when referring to the MCS event

Line 95 – Change "descend" to "descends"

Line 120 – Should read "…by the Institute of Atmospheric…

Line 123 – Avoid starting a sentence with O3. Spell this out instead.

Line 128- You discuss numerous acronyms here without defining them beforehand – this includes AIRS, EOS, OMI, etc.

Line 170 – Change to "guarantees"

Line 171 – Change to "To explicitly…"

Line 180  - Change to "Covering the entire lifespan of the MCS…"

Line 190 – Change "…would be described…" to "…is described…"

Line 201 – What does the squiggly line mean between 45 and 50? Is this a range? If so, change this to a dash.

Lines 204-205 – Change "….normal diurnal cycles since…" to "…subsequent diurnal cycles…"

Line 210 – Use a better word than "severer"

Line 224 – Change "…and MCS took place…" to "…during the MCS event…"

Line 225 – Change to "…in cities such as…"

Lines 234-235 – I suggest rewriting this sentence. Using words like "well revealed" or "averagely" sound awkward.

Line 237 – Change to "the SHATLE…"

Line 237 – Define the acronym SHATLE for the reader here.

Line 244 – Change "continuously" to "continuous"

Line 245 – Change to "…in the next 10 minutes and remained high for the next eight hours."

Line 270 – remove "originated"

Line 273 – "typhoon" should be plural

Line 277 – Change "While in this case study…" to something simpler like "In this study…"

Line 278 – Refer to the storm as "Typhoon In-fa"

Line 293 – Change to "…even though In-fa was in its dissipation stage."

Line 303 – refer to "suggest" in the present tense

Line 308-311 – Break this up into more than one sentence.

Line 312-315 – Break this up into more than one sentence.

Line 345 – Change "descending" to "descent"

Line 346 – Refer to dates with the proper format

Line 348 – Change "…located at southwest…" to "…located southwest…"

Line 350 – Change to "…evolved into a bow-echo MCS…."

Lines 354-356 – Clean this sentence up. I do not understand what you are trying to say.

Line 368 – Change "…that located…" to just "located"

Line 375 – Change "…of thunderstorm…" to "…of a thunderstorm…"

Line 382 – Change "descending" to "descent"

Line 387 – Explain what you mean by overforecasted convection distribution. Does this mean the model was too aggressive in developing convection? If so, state this more clearly.

Line 387 – Change "….well captures…" to "…does a good job capturing…"

Figure 9 – Change caption from "…occurred in the night…" to "…occurred during the night…"

Line 402 – Change "descending" to "descent"

Line 402-405 – Clean this entire sentence up. It is difficult to read.

Line 420 – Change "upper" to "up"

Line 423 – Change "descending" to "descent"

Line 436 – Change "dissipating" to "dissipation"

Line 445 – Change "perturbated" to "perturbed"

Line 446 – Change "pre-existed" to "pre-existing"

Line 460 – Remove "relatively"

Line 480 – Change "perturbate" to "perturb"

Line 483 – Change "descending" to "descent"

Line 487 – What does "at the midnight" mean?

Line 488 – Change "pre-existed" to "pre-existing"

Line 491 – Replace "rapider" with a better word

Line 500-502 - Clean this sentence up. It reads awkwardly.

Line 503 – Change "threatens" to "threats"

Line 505 – Change "…in the mechanisms…." to "…of the mechanisms…"

Line 689 – The Preston reference should list "Barth" as a coauthor (not Berth)

Figure 5 – The second sentence of this caption needs to be fixed – "…the last six track information…" does not make sense.

Figure 7 – Include the year of the dates with this caption

Figure 8 – Remove "occurred" in the caption

---

## Author Comment (AC1)

Chen et al. reported a comprehensive analysis of the abrupt increase of surface ozone observed in the North China Plain on the night of July 31 2021. The authors employed various datasets and tools in their analysis, including (1) the air pollutant observational data from the national monitoring stations, (2) high-frequency ground-based observational data during a campaign in this region, (3) vertical profile observation of ozone, (4) radiosonde meteorological data, (5) reanalysis data, (6) regional meteorological model, and (7) back-trajectory model. They found out that the sudden increase of surface ozone was not due to horizontal transport; instead, the dying typhoon and a local mesoscale convective system brought the high-ozone/low-CO air to the lower troposphere and even the ground surface. The authors did a commendable job by applying various tools to analyze a unique atmospheric phenomenon that has implications for air quality management. I support acceptance of the paper once the following concerns are addressed.

Thank you very much for your helpful suggestions which help us improve our manuscript substantially. We have modified our manuscript according to the comments.

**General comment:**

1.  The authors emphasized that this paper is about the effect of a "dying" typhoon and "shallow" convection in the title, abstract, and many places in the main texts. While this is correct based on the authors' analysis, I wonder would it be better to generalize the mechanism? If I understand correctly, a typhoon (dying or not) likely causes stratospheric ozone that brings the stratospheric ozone to the upper and middle troposphere, and a follow-up mesoscale convective system (shallow or not) would then transport the high-ozone/low-CO air further down to the lower troposphere.

Several researchers have found that a large portion of stratospheric intrusions reach the top of planetary boundary layer (PBL), while only a few of them could penetrate

into the ground surface (e.g., Ott et al., 2016; Trickl et al., 2020). This study addressed a coincidence of conditions for direct stratospheric intrusion to the surface including the large-scale typhoon and the meso-scale convective systems (MCSs). At first, the dying typhoon induced stratospheric intrusion into the middle-to-low troposphere, and then the MCSs facilitated the transport pathway of stratospheric ozone-rich air to contact the surface, and hence affected the surface air quality.

Ott, L. E., Duncan, B. N., Thompson, A. M., Diskin, G., Fasnacht, Z., Langford, A. O., Lin, M., Molod, A. M., Nielsen, J. E., Pusede, S. E., Wargan, K., Weinheimer, A. J., and Yoshida, Y.: Frequency and impact of summertime stratospheric intrusions over Maryland during DISCOVER-AQ (2011): New evidence from NASA's GEOS-5 simulations, J. Geophys. Res., 121, 3687–3706, https://doi.org/10.1002/2015JD024052, 2016.

Trickl, T., Vogelmann, H., Ries, L., and Sprenger, M.: Very high stratospheric influence observed in the free troposphere over the northern Alps - just a local phenomenon?, Atmos. Chem. Phys., 20, 243–266, https://doi.org/10.5194/acp-20-243-2020, 2020.

2.      Meng et al. (2022) reported a very similar process (an anomaly in surface ozone due to the passing typhoon) in the same region (NCP). I am aware that the authors of the present work started their analysis before this recent paper (Meng et al., 2022) was published, but it would be beneficial to the readers if the authors could add some relevant discussions.

Thank you for your suggestion and the reference. We also agree that convective systems including typhoons can induce significant stratospheric intrusion and alter the meteorological conditions necessary for ozone production and accumulation. Case study and statistical analysis of typhoon's impacts on ozone variations would improve our understandings of the role of such synoptical convective systems played in ozone budget and air quality. The reference paper has been added in the manuscript.

3.      The designs of the manuscript and figures require some improvements:

(1)      Section 2.3, please add a figure (at least in the supplement) showing the domain setting of the WRF simulation. I would like to know whether the inner domain covers the region with strong vertical transport.

Thank you for your suggestion, the domain coverage used in the simulation is shown in Figure S1 in the supplementary file.

(2)      In The paragraph starting at line 174, a table listing all WRF model parameterizations and setups will be very clear.

Thank you for pointing this out, the table of basic WRF settings is added in Table 1 (Section 2.3).

(3)      Line 191, why not describe the setup of FLEXPART-WRF here? How many simulations? What is the location (lat, long, and altitude) of particle release? How many days for each simulation?

Thank you for pointing this out. The FLEXPART is driven by meteorology field from high-resolution WRF model output. The locations of backward trajectory simulations were set to Binzhou and Qingdao city between 1000-950 hPa at 04:00 LST on 1 August (Line 412-417). The time length of simulations was 28 hour.

(4)      Line 194-197, these two sentences seem out of scope. Consider removing. I don't think it is necessary to mention the increase in ozone in the past decade in this region.

Thank you for your suggestion. We have removed these sentences.

(5)      Line 208-210, this sentence is a bit odd too. Consider removing.

Thank you for your suggestion. We have removed this sentence.

(6)     Figure 2, showing the "departure from the 10-day averaged ozone", is not a good choice to demonstrate the sudden increase of surface ozone. Instead, I believe Figure S2 (with the average diurnal pattern of ozone) is a much better option for showing the anomaly of surface ozone in this region. Similarly, I would recommend drawing a similar figure of CO to replace the original Figure 3.

Thank you. The variations of ozone concentrations are compared with their 10-day averaged value during which the NCP was influenced by Typhoon In-fa. It is rather clear to show the "weak" and "strong" phases of ozone variations as In-fa travelled through the NCP and emphasize the role of typhoon in influencing ozone concentrations.

(7)     If possible, Figure 4 should also be replaced with one similar to the original Figure S2. In fact, in line 249, the authors stated that "Compared with the normal nighttime ozone concentrations (an average of 36.6 ppbv), the magnitudes of surface ozone surge due to stratospheric intrusions were approximately 40-50 ppbv". If the "normal nighttime ozone concentrations" were already shown in Figure 4, i.e., with the average diurnal pattern, readers would easily see the "departure" of ozone/CO from their "normal nighttime concentration". Also, I suggest only including the "hour" in the X-axis should be informative enough.

Thank you for your points. Figure 4 shows the high-frequency observations of ozone and CO during the night of 31 July 2021, which are more valuable and evident to track the exact time when stratospheric ozone-rich and CO-poor air reached the ground. For the sake of consistency, we kept the original labels of X-axis.

(8)     Figure 5, what do the positive/negative vertical velocities represent? Positive values (blue) are winds going down to the surface? Or the other way around? Please clarify in the figure caption.

Yes, the positive vertical velocity (omega) in the unit of Pa $s^{-1}$ represents downward air motions and the negative one represents the upward air motions. Thanks for your suggestion. We now add the information in the figure caption.

(9)     In Figure 7, similarly, I don't understand why an average level of ozone between surface and 700hPa is used as a baseline. Shouldn't the baseline be the 10-day average vertical profile? With the current figure, the readers must be puzzled why the surface ozone concentrations on July 31 and Aug 1 are lower than the average, while the other sections repeatedly show that the surface ozone concentrations in NCP are larger than the average.

According to previous studies, only a few percent of the ozone at about 700 hPa could be clearly attributed to direct stratospheric intrusions (Elbern et al., 1997; Stohl et al., 2000). For this reason, the 10-day dewpoint depressions (humidity information; Fig. 6) and ozone concentrations below 700 hPa are averaged as the baseline to track the descending stratospheric dry and ozone-rich air from upper levels. Since the ozone baseline here is an average between the surface and 700 hPa, as a result, the ozone concentration at the surface level was lower than the baseline value.

Elbern, H., J. Kowol, R. Sladkovic, and A. Ebel, Deep stratospheric intrusions: A statistical assessment with model guided analyses, Atmos. Environ., 31, 3207 – 3226, 1997.

Stohl, A., et al., The influence of stratospheric intrusions on alpine ozone concentrations, Atmos. Environ., 34, 1323 – 1354, 2000.

(10)     Line 360-364, this information should be moved before mentioning Bow-echoes.

Thank you for this valuable suggestion. The description of bow-echo MCSs was moved to Line 358-363.

(11)  Figure 8, why not show the ozone data at all sites at all times and use a colour scale that covers 10 to 100 ppbv? With the current layout, there is no way to tell how much ozone is increased from 2100LST to 0100LST at stations like JN/BZ. It could just be 1 ppbv of increase (if ozone at JN/BZ were 79 ppbv at 2100LST and 80 ppbv at 0100LST) or >80 ppbv of increase (if ozone were <1 ppbv at 2100LST and >80ppbv at 0100LST).

Sorry for this. We have tried to plot all the ozone observations ranging from 10 to 120 ppbv. However, the radar reflectivity field is covered when more ozone data are mapped, and hence it is hard to tell the fine-scale structure of MCSs. For this reason, only ozone observations with high concentrations (80-110 ppbv) are shown in Figure 8, and stations with ozone concentrations less than 80 ppbv are not displayed for clearness.

**Specific comment**:

4.  line 37, this line reads like both "water vapour" and "carbon monoxide" are primarily emitted from combustion processes, while only CO is. Consider revising it.

Thank you for your suggestion, the sentence has been modified.

5.  Line 75, what problems "require in-depth investigation"?

Thank you, we have modified the sentence.

6.  Line 199, is this 36.6 ppbv calculated in this study or from a previous study?

The nighttime averaged ozone concentration in the 2021 summer over NCP was calculated based on observations. We have added this information in this revision.

7.  Line 228-230, this sentence sounds important. Any figures/data to support it?

Yes, the spatial coverage can be inferred from the cities with increasing ozone and decreasing CO as shown in Figure 2 and Figure 3. Regions associated with high

nighttime ozone concentrations due to stratospheric intrusion are also evident in Figure 8.

8.     Line 237, "filed" should be "field". I have spotted a few more typos. Please check through the manuscript.

Thank you for pointing this out. We have corrected typos throughout the manuscript.

9.     Line 239, it is >30% increase from 45 to 60 ppbv. I would not call it "slightly higher".

Thank you for pointing this out, we have modified the sentence.

10.     Line 250-251, an increase in ozone from ~36 ppbv to ~80 ppbv is a large enhancement, but this level of ozone (80 ppbv) should be very common in this region. I suggest toning down the phrase "great threats".

Thank you for pointing this out, we have modified the sentence.

11.     Line 262, somehow the authors missed "anthropogenic emission"?

Thank you. We added "anthropogenic emissions" in the sentence.

12.     Line 266-267, this is probably true, but it will be better if the evidence is presented.

Thank you. Here we just provide the results of previous studies concerning the convection and STE.

13.     Line 271, "indicate" should be "indicated".

Corrected in Line 274.

14.     Line 417, "the analysed at detail" should be "be analysed in detail".

Corrected in Line 427.

Reference:

Meng, K., Zhao, T., Xu, X., Hu, Y., Zhao, Y., Zhang, L., Pang, Y., Ma, X., Bai, Y., Zhao, Y. and Zhen, S., 2022. Anomalous surface $O_3$ changes in North China Plain during the northwestward movement of a landing typhoon. Science of The Total Environment, p.153196. http://dx.doi.org/10.1016/j.scitotenv.2022.153196

Thank you for providing this reference, which was added in the manuscript.

---

## Author Comment (AC2)

**#Reviewer 2**

**General comment**:

This is an interesting case study of significant stratospheric ozone transport down to the Earth's surface by a dying typhoon, affecting local and regional air quality. Several observational data and modeling tools are applied to analyse/confirm the downward transport of ozone with stratospheric origin. Overall, this is a well-designed study which is relatively easy to follow. Such events of direct ozone transport have implications for air quality, contributing in ozone standards exceedances. The paper fits well within the scope of ACP and I recommend publication after the following comments are addressed.

We thank for the reviewer's helpful comments. We have revised the manuscript thoroughly according to the comments and the manuscript has been improved substantially. The point-to-point responses are listed below.

**Comments**:

The only thing I found missing in the analysis are humidity measurements near the surface from ground-based meteorological stations. This would offer the temporal variability of humidity near the surface, likely supporting the case that the observed ozone increases are of stratospheric origin. Is this feasible?

Yes, humidity observations near the surface would provide extra evidences for the stratospheric intrusion. However, it is a pity that we have only limited humidity data with coarse temporal resolution from ground-based meteorological stations to explicitly track the moisture variations.

L41-42: A more scientific definition of the tropopause is rather necessary here.

Thank you, we have modified the sentence.

L59: Some additional references of direct SI impact on surface ozone concentrations are needed here like Akritidis et al. (2010), Dreessen (2017), and Knowland et al. (2017).

Thank you for providing these references. These references are added in this revision.

L60: Meul et al. (2018) and Akritidis et al. (2019) also suggested an increase of STT in a future climate.

Thank you for providing these references. We have added these references in this revision.

L74-76: Maybe some references are needed here. Which are the fundamental problems requiring in-depth investigation?

Thank you for your suggestion, some references were added in the manuscript. We have modified the sentence.

L78-79: "the stratospheric ozone-rich airmass was transported downward to the surface". This is Introduction and such statements are not yet supported. I suggest removing or rephrase.

Thank you, we have modified the sentence.

L137-138: "along with ground-based automatic weather station observations": Which exactly? Do you mean the radar data? If not, are these shown anywhere in the paper?

Sorry for this mistake, we have modified the expression.

L298: Why is the PV = 2.5 pvu isosurface selected for tropopause representations? Usually, 2 and 1.5 pvu are used. A reference/rationale for that selection would be helpful.

Thank you for point out this problem. Values of potential vorticity ranging between 1-4 PVU can be found in the literature. For example, Hoerling et al. (1991) chose a 3.5-PVU isoline for their tropopause analysis because it statistically agreed with the thermal tropopause. In the study of Maddox and Mullendore (2018), they found a 2.89-PVU threshold was best fit the constant-altitude tropopause. In this study, we applied the 2.5-PVU isosurface to represent the dynamical tropopause following Wirth (2003). The reference is added in the manuscript.

Wirth, V.: Static stability in the extratropical tropopause region, J. Atmos. Sci., 60, 1395–1409, https://doi.org/10.1175/1520-0469(2003)060<1395:SSITET>2.0.CO;2, 2003.

Figure 4: Vertical lines delimiting the O3 increase and CO decrease (similar to Figure 3) would be helpful here.

Thank you for this suggestion, the lines were added in Figure 4.

Figure 7: Here the 10-day average "between the surface and 700 hPa" is used as baseline for the $O_3$ profiles. What is the rationale behind this selection (between the surface and 700 hPa)? As $O_3$ increases in general with height, I think it is likely that the positive (red) departures in the troposphere are partially normal, masking the STT effect.

Thank you for pointing out this problem. According to previous studies, only a few percent of the ozone at about 700 hPa could be clearly attributed to direct stratospheric intrusions (Elbern et al., 1997; Stohl et al., 2000). For this reason, the 10-day dewpoint depressions (humidity information; Fig. 6) and ozone concentrations below 700 hPa are averaged to represent the features of low-level air. And the averaged values of dewpoint depressions and ozone concentrations were used as the baseline to track the descending stratospheric dry and ozone-rich air from upper levels. Through day-to-day comparisons, it is clearly that large positive departures extended from upper troposphere to the lower altitudes (Fig. 7a-c).

Elbern, H., J. Kowol, R. Sladkovic, and A. Ebel, Deep stratospheric intrusions: A statistical assessment with model guided analyses, Atmos. Environ., 31, 3207 – 3226, 1997.

Stohl, A., et al., The influence of stratospheric intrusions on alpine ozone concentrations, Atmos. Environ., 34, 1323 – 1354, 2000.

L407-408: "when the stratospheric airmass had reached the surface". Where does this arise from? If it's based on a Figure, please include it in parentheses e.g. (see Fig. 2)

Thank you for pointing this out. Based on Fig. 2 and Fig. 4, it is confirmed that the stratospheric ozone-rich air had reached the surface. We have modified the sentences in Line 416.

Figure 11: What do the magenta contour line labels describe? Is this percentage (%) of total number release? Please include this information in the respective caption.

Sorry for this, the magenta contour lines represent the number of tracers. We have added more descriptions about the contour lines in the caption (Line 461).

**Technical comments**

L16: "on 31 July 1 to 6:00" delete "1"

Thank you. Corrected and checked throughout the manuscript.

L25: Please move FLEXPART and WRF full names in the previous line where are referred.

Thank you. We have modified the sentence in Line 23-25.

L34: and the troposphere

Corrected.

L34: atmospheric composition

Corrected.

L56: STT usually stands for Stratosphere-to-Troposphere Transport which is not the case here. Please remove STT or change the phrase.

Removed.

L76: origin

Corrected throughout the paper.

L80: Compare with -> Compared to

Thank you. Corrected.

L87: Since here you are referring to a specific study I suggest to directly mention it. "Chen et al. (2021) evaluating the impacts of typhoons on tropospheric ozone showed..".

Thank you for your suggestion. We have modified the sentence in Line 87-88.

L134: and they show -> showing

Modified.

L151: the stratospheric dryness -> dry stratospheric air

Modified.

L199: Please include nighttime definition (hour range).

Thank you for your suggestion, the nighttime is defined as 19:00-06:00 LST and has been added in the manuscript (Line 198).

L228: "confirms" is somehow strong here, I suggest "supports the case"

Corrected in the manuscript (Line 227), and thank you for this valuable suggestion.

L357: lasting->lasted

Corrected.

L458: "occurring at nigh". As this is the beginning of the conclusions, the date of occurrence should be also stated.

Thank you for this valuable suggestion, we have added the date in the manuscript (Line 467).

L459-460: "while the impacts of stratospheric intrusions on surface ozone are relatively less studied". This is somehow not connected to the previous part of the sentence, thus, I suggest to split in two sentences.

Thank you for your suggestion, we have modified the sentence (Line 468).

L512-513: "which underscores the necessity of considering these processes into the global model of atmospheric chemistry." This phrase is somehow strange. What do you mean by global model of atmospheric chemistry? Please rephrase.

Sorry for this, we have modified the sentence (Line 524).

References

Akritidis, D., Zanis, P., Pytharoulis, I. et al. A deep stratospheric intrusion event down to the earth's surface of the megacity of Athens. Meteorol Atmos Phys 109, 9–18 (2010). https://doi.org/10.1007/s00703-010-0096-6

Akritidis, D., Pozzer, A., and Zanis, P.: On the impact of future climate change on tropopause folds and tropospheric ozone, Atmos. Chem. Phys., 19, 14387–14401, https://doi.org/10.5194/acp-19-14387-2019, 2019.

Dreessen, J. (2019). A Sea Level Stratospheric Ozone Intrusion Event Induced within a Thunderstorm Gust Front, Bulletin of the American Meteorological Society, 100(7), 1259-1275.

Knowland, K. E., Ott, L. E., Duncan, B. N., & Wargan, K. (2017). Stratospheric intrusion-influenced ozone air quality exceedances investigated in the NASA MERRA-2 reanalysis. Geophysical Research Letters, 44, 10,691– 10,701. https://doi.org/10.1002/2017GL074532

Meul, S., Langematz, U., Kröger, P., Oberländer-Hayn, S., and Jöckel, P.: Future changes in the stratosphere-to-troposphere ozone mass flux and the contribution from climate change and ozone recovery, Atmos. Chem. Phys., 18, 7721–7738, https://doi.org/10.5194/acp-18-7721-2018, 2018.

Thank you for providing these references. We have added the references in this revision.

---

## Author Comment (AC3)

**#Reviewer 3**

**General Comments**

Overall, this is an interesting study that adds insight into the role that typhoons and mesoscale convective systems play in transporting stratospheric ozone to the surface. The science of this study is sound, the paper is well organized, and the figures nicely complement the text. However, there is poor grammar throughout the paper which makes it difficult to read at times. I include specific comments below that address many of these grammatical errors. My comments below are mainly minor in scope. As a result, I fully support acceptance of this paper once the following comments are addressed. I think this paper will be a good addition to the ACP journal.

We appreciate the reviewer's time and efforts to improve the language and quality of the paper. We have revised the manuscript thoroughly according to the comments and the manuscript has been improved substantially. The point-to-point responses are listed below.

**General Comments**

You should add a table that lists basic parameters you used for your WRF simulation. This is standard practice when running a model simulation.

Thank you for pointing this out. A table of basic WRF settings has been added in Section 2.3

You should capitalize "Typhoon" since you are referring to a specific one by name. You should do this throughout the paper anytime you reference "Typhoon In-fa".

Thank you very much. Corrected and checked throughout the manuscript.

Be consistent with how you refer to water "vapor". You go back and forth between "vapor" and "vapour" throughout the paper. These are both correct so select one and stick with it.

Thank you for pointing out this problem, we have corrected these mistakes throughout the manuscript.

Make sure to include articles like "a" and "the" before referring to a singular noun. I did my best to address some of these specific instances below but I did not discuss all of them below.

Thank you so much. We appreciate your kindness and efforts to improve the quality of this paper.

Spell out "Figure" if it begins a sentence.

Thank you so much. We learned a lot from your suggestions.

**Specific Comments**

Line 14 – Change "…at night…" to "…during the night…"

Corrected.

Line 16 – Provide the time units (LT or UTC) for 23:00 and 6:00

Added in the manuscript.

Line 16 – Change "01 August, 2021" to "1 August 2021". There are other instances of this throughout the paper so fix date formats accordingly.

Corrected throughout the manuscript.

Line 19 – Change "…of…" to "…of a…"

Corrected.

**Lines 20-21 – Reword this sentence. You make a good point, but this is lost here because of grammar issues.**

Thank you for your suggestion, we have modified the sentence.

Lines 23,66 – You should define the MCS acronym as MCSs because you refer to "systems" which is plural. Once this is defined then you can use "MCS" in the future when referring to just a single mesoscale convective system.

Thank you for point out this problem. We have modified them throughout the manuscript.

Line 26 – Change to "…still bring down a stratospheric…"

Corrected throughout the manuscript.

Line 28 – Change "pre-existed" to "pre-existing"

Corrected throughout the manuscript.

Line 29 – Change "descending" to "descent"

Corrected throughout the manuscript.

Line 39 – Change to "…contains only 5-10% of atmospheric ozone, as well as high water vapor and CO concentrations due…"

Corrected.

Line 43 – Be careful here. You refer to deep convection which is singular but then you use"they" later in the sentence. Be consistent with your grammar.

Thank you for pointing this out. We have modified the sentence.

Line 48 – Change to "…in the UTLS."

Modified.

Line 51 – Change to "…from the ABL to the UTLS…"

Modified.

Line 53 – Change to "…for the radiation balance…"

Corrected.

Line 58-62 – This sentence is too long. I suggest breaking this up into two separate sentences.

Thank you for your suggestion, we have modified the long sentence in the manuscript.

Line 80 – Change "Compare" to "Compared" since the comparison occurred in the past tense.

Corrected.

Line 82 – Change to "Such a significant…"

Corrected.

Line 85 – Change "…in 2021 summer…" to "…in Summer 2021."

Corrected.

Line 90 – Replace "unneglectable" with a better word here

We changed "unneglectable" to "substantial" in the manuscript.

Line 91 – Use a singular verb "was" instead of "were" when referring to the MCS event

Modified in the manuscript.

Line 95 – Change "descend" to "descends"

Corrected.

Line 120 – Should read "…by the Institute of Atmospheric…

Corrected.

Line 123 – Avoid starting a sentence with O3. Spell this out instead.

Modified.

Line 128- You discuss numerous acronyms here without defining them beforehand – this includes AIRS, EOS, OMI, etc.

Sorry for these mistakes. We have added the definitions of theses acronyms.

Line 170 – Change to "guarantees"

Corrected.

Line 171 – Change to "To explicitly…"

Corrected.

Line 180 - Change to "Covering the entire lifespan of the MCS…"

Corrected.

Line 190 – Change "…would be described…" to "…is described…"

Corrected.

Line 201 – What does the squiggly line mean between 45 and 50? Is this a range? If so, change this to a dash.

Corrected.

Lines 204-205 – Change "….normal diurnal cycles since…" to "…subsequent diurnal cycles…"

Corrected.

Line 210 – Use a better word than "severer"

Thank you for this suggestion, we have deleted this sentence according to the reviewer's suggestion.

Line 224 – Change "…and MCS took place…" to "…during the MCS event…"

Modified.

Line 225 – Change to "…in cities such as…"

Modified.

Lines 234-235 – I suggest rewriting this sentence. Using words like "well revealed" or"averagely" sound awkward.

Thank you for your suggestion. We have rephrased the sentence.

Line 237 – Change to "the SHATLE…"

Corrected throughout the manuscript.

Line 237 – Define the acronym SHATLE for the reader here.

Sorry for this, the acronym SHATLE was defined in Line 120.

Line 244 – Change "continuously" to "continuous"

Corrected.

Line 245 – Change to "…in the next 10 minutes and remained high for the next eight hours."

Corrected.

Line 270 – remove "originated"

Corrected.

Line 273 – "typhoon" should be plural

Changed, and checked throughout the manuscript.

Line 277 – Change "While in this case study…" to something simpler like "In this study…"

Modified.

Line 278 – Refer to the storm as "Typhoon In-fa"

Thank you, we have modified the expression throughout the manuscript.

Line 293 – Change to "…even though In-fa was in its dissipation stage."

Thank you, we have modified the expression throughout the manuscript.

Line 303 – refer to "suggest" in the present tense

Corrected.

Line 308-311 – Break this up into more than one sentence.

Thank you for your suggestion. We have modified the long sentence.

Line 312-315 – Break this up into more than one sentence.

Thank you. We have modified the long sentence.

Line 345 – Change "descending" to "descent"

Corrected and checked throuhout the manuscript.

Line 346 – Refer to dates with the proper format

Thank you. Corrected and checked throuhout the manuscript.

Line 348 – Change "…located at southwest…" to "…located southwest…"

Corrected.

Line 350 – Change to "…evolved into a bow-echo MCS…."

Corrected.

Lines 354-356 – Clean this sentence up. I do not understand what you are trying to say.

Sorry for this. We have modified the expression.

Line 368 – Change "…that located…" to just "located"

Corrected.

Line 375 – Change "…of thunderstorm…" to "…of a thunderstorm…"

Thank you. Modified in the manuscript.

Line 382 – Change "descending" to "descent"

Corrected.

Line 387 – Explain what you mean by overforecasted convection distribution. Does this mean the model was too aggressive in developing convection? If so, state this more clearly.

Thank you. Compared with radar observations, the simulated convection distributes over larger regions, and hence we said WRF overforecastes the convection coverage in this case. Modified in Line 393-396.

Line 387 – Change "….well captures…" to "…does a good job capturing…"

Modified.

Figure 9 – Change caption from "…occurred in the night…" to "…occurred during the night…"

Corrected.

Line 402 – Change "descending" to "descent"

Corrected. Thank you.

Line 402-405 – Clean this entire sentence up. It is difficult to read.

Sorry for this. We have rephrased the long sentence.

Line 420 – Change "upper" to "up"

Corrected.

Line 423 – Change "descending" to "descent"

Corrected.

Line 436 – Change "dissipating" to "dissipation"

Corrected.

Line 445 – Change "perturbated" to "perturbed"

Corrected.

Line 446 – Change "pre-existed" to "pre-existing"

Thank you. Corrected.

Line 460 – Remove "relatively"

Removed.

Line 480 – Change "perturbate" to "perturb"

Corrected.

Line 483 – Change "descending" to "descent"

Corrected. Thank you.

Line 487 – What does "at the midnight" mean?

We changed "at the midnight" to "during the night". Thank you for pointing this out.

Line 488 – Change "pre-existed" to "pre-existing"

Corrected.

Line 491 – Replace "rapider" with a better word

Thank you. We changed "rapider" to "faster".

Line 500-502 - Clean this sentence up. It reads awkwardly.

Sorry for this. We have rephrased the long sentence.

Line 503 – Change "threatens" to "threats"

Corrected.

Line 505 – Change "…in the mechanisms…." to "…of the mechanisms…"

Corrected.

Line 689 – The Preston reference should list "Barth" as a coauthor (not Berth)

We are very sorry for this mistake. Corrected in the reference.

Figure 5 – The second sentence of this caption needs to be fixed – "…the last six track information…" does not make sense.

Thank you for pointing this out. The magenta crosses represent the tracks of Typhoon In-fa during its dissipation stage with a time interval of 6 hours. We have modified the caption.

Figure 7 – Include the year of the dates with this caption.

Added in the caption, thank you for pointing this out.

Figure 8 – Remove "occurred" in the caption

Removed.

---

## Author Comment (AC4)

**#Reviewer 4**

This paper discussed the downward transport of stratospheric ozone to the troposphere as well as down to the surface through a combined effect of a dying typhoon In-fa and shallow local mesoscale convective system (MCS). They analyzed the ozone and CO concentration, meteorological reanalysis data, radiosonde data, and FLEXPART-WRF simulation. The downward transport of stratospheric ozone-rich air to the surface will degrade surface air quality and affect human health. Overall, the paper is good. It studied an important topic, used various observations. However, it still has some major weak points.

We thank the reviewer for the positive comments and very careful reading of our article. The corrections are addressed below.

**General comments:**

1.    Because the downward transport was caused by typhoon In-fa, it would be nice to have a brief introduction of typhoon In-fa in section 2. Please include a plot showing the development of the typhoon In-fa (e.g., radar reflectivity for different times), and a plot showing the path of the typhoon In-fa. This will help the reader to understand the discussion of the second part.

The development of typhoon In-fa was described in Line 280-298; and the track information of In-fa was provided in Fig. 5a (magenta cross symbols). Thank you for your suggestion, we downloaded the radar reflectivity maps of In-fa from 27-30 July 2021 as shown in Fig. S5 in the supplementary file.

2.    Lightning-generated NOx could also increase downwind ozone level. The paper did not prove that the ozone increase is not caused by LNOx generated by previous storms.

Yes, the precursors (NOx and VOCs) of ozone can significantly influence the

ozone concentrations, however, as stated in the manuscript, this ozone surge event occurred in the nighttime when photochemical reactions ceased. In the meantime, the storm was weakly electrified and hence showed low lightning flash rates (Line 383).

3. In the paper, they calculate the 10-day mean O3/CO as the baseline. However, the 10-day mean included the days affected by typhoon In-fa. Therefore, is hard to tell what's the normal condition. It might be better to use the 10-day mean before the typhoon period as the baseline.

Thank you for this suggestion. We calculated the 10-day mean values of ozone and CO covering the typhoon and post-typhoon period in order to show the "weak" and "strong" phases of ozone more clearly under the influence of In-fa.

4. In this paper, they run WRF with tracer instead of using WRF-Chem. However, LNOx and other ozone precursors could also affect the results. Please explain why you choose not to use WRF-Chem or other chemistry models. The ozone production is not significant in the first few hours, however, previous studies found that there would be a great ozone increase in the downwind side on the next day. If you insist to use WRF with tracers, you need to convince the reader that your results would not be affected by any ozone chemistry reactions.

Thank you for pointing this out. Because this ozone surge event exactly occurred at night when sunshine was not available and hence photochemical reactions ceased, we focused on the dynamical transport of ozone (stratospheric air here) using the WRF model with tracers. To better resolve the vertical structure of the storm, the vertical spacing was increased with a resolution of ~200 m.

**Specific comments:**

Line 44, here are some references for deep convective transport of surface pollution and ozone precursors to upper troposphere:

Dickerson, R. R., Huffman, G. J., Luke, W. T., Nunnermacker, L. J., Pickering, K. E., Leslie, A. C. D., Lindsey, C. G., Slinn, W. G. N., Kelly, T. J., Daum, P. H., Delany, A. C., Greenberg, J. P., Zimmerman, P. R., Boatman, J. G., Ray, J. D., and Stedman, D. H. (1987). Thunderstorms: an important mechanism in the transport of air pollutants, Science, 235:460-465.

Pickering, K.E., Thompson, A.M., Scala, J.R., Tao, W.-K., Simpson, J., and Garstang, M. (1991). Photochemical ozone production in tropical squall line convection during NASA Global Tropospheric Experiment/Amazon Boundary Layer Experiment 2A. J. Geophys. Res. 96, 3099–3114.

Pickering, K.E., Thompson, A.M., Scala, J.R., Tao, W.-K., and Simpson, J. (1992c). Ozone production potential following convective redistribution of biomass burning emissions. J Atmos Chem 14, 297–313.

Li, Y., Pickering, K.E., Allen, D.J., Barth, M.C., Bela, M.M., Cummings, K.A., Carey, L.D., et al. (2017). Evaluation of deep convective transport in storms from different convective regimes during the DC3 field campaign using WRF-Chem with lightning data assimilation. J. Geophys. Res. Atmos. 122, 2017JD026461.

Thank you, the above references have been added in the manuscript.

Line 110, it would be nice to have a brief introduction of typhoon In-fa in section 2. Please include a plot showing the development of the typhoon In-fa (e.g., radar reflectivity for different times), and a plot showing the path of the typhoon In-fa. This will help the reader to understand the discussion of the second part.

Thank you, more information of typhoon In-fa has been added in the manuscript and supplementary file (Fig. S5).

Line 166, please include the reference for WRF.

Thank you. The reference for WRF was added in the manuscript.

Skamarock, W.C., Klemp, J.B., Dudhia, J., Gill, D.O., Barker, D.M., G Duda, M., Huang, X.-Y., Wang, W., and Powers, J.G.: A description of the advanced research WRF

version 3. NCAR Tech. Note NCAR/TN-475þSTR, p. 113. https://doi.org/10.5065/D68S4MVH, 2008.

Line 167, please add a figure showing the location of each domain in supporting information.

Thank you for this suggestion. The simulation domains were shown in Fig. S1.

Line 185, why do you choose WRF instead of WRF-Chem? See general comments 4.

Thank you, because this nighttime ozone surge case occurred when photochemical reactions ceased, we used WRF with tracers to study the dynamical transport pathways of stratospheric ozone-rich air.

Figure 2, see general comments 3.

Thank you, the averaged values of ozone and CO under the influence of typhoon In-fa were calculated and used as the baseline in order to show the "weak" and "strong" phase of ozone for different times

Figure 2, please add a map showing the storm location during the ozone surge period.

Thank you, the storm location during the ozone surge period was provided in Fig. 8 in the manuscript.

Line 226, CO is also an important tracer for deep convective transport. Please include references here. "CO is offen….(add references)"

Thank you. The reference for CO was added in the manuscript.

Pochanart, P., Akimoto, H., Kajii, Y., and Sukasem, P.: Carbon monoxide, regional-scale, and biomass burning in tropical continental Southeast Asia: Observations in rural Thailand. J. Geophys. Res.-Atmos., 108, 4552, https://doi.org/10.1029/2002JD003360, 2003.

Lin, Y.-C., Hsu, S.-C., Lin, C.-Y., Lin, S.-H., Huang, Y.-T., Chang, Y., and Zhang,

Y.-L.: Enhancements of airborne particulate arsenic over the subtropical free troposphere: impact of southern Asian biomass burning, Atmos. Chem. Phys., 18, 13865–13879, https://doi.org/10.5194/acp-18-13865-2018, 2018.

Line 250, please mention the ozone exceedance level, and compare the observed ozone level to the ozone exceedance level. Otherwise, you cannot conclude that "which can pose great threats to human health…"

Thank you, the Chinese National Ambient Air Quality Standard for ozone exceedance level is 82 ppbv (Li et al., 2020). We added this exceedance level in the manuscript.

Li, K., Jacob, D. J., Shen, L., Lu, X., De Smedt, I., and Liao, H.: Increases in surface ozone pollution in China from 2013 to 2019: anthropogenic and meteorological influences, Atmos. Chem. Phys., 20, 11423–11433, https://doi.org/10.5194/acp-20-11423-2020, 2020.

Line 268, please explain more about "no influence from ozone precursors from biomass burning or LNOx". See general comments 2.

Thank you. Because the photochemical reactions had ceased when the nighttime ozone surge event occurred, and there were synthetic decreases of CO concentrations, suggesting that there were few influences from biomass burning. In terms of lightning activities, the storm was weakly electrified and produce low lightning flash rate. A total of 362 cloud-to-ground lightning flashes were detected from 21:00 LST on 31 July to 06:00 LST on 1 August 2021 within 50-km radius of Zhanhua station. For the reason above, we concluded that there was no significant influence from ozone precursors from biomass burning or LNOx.

Figure 5, please label time in each plot.

Thank you for pointing this out. The time label is expressed as "Month/Day/Hour" in the bottom-right corner of each panel.

Line 400, could you add a forward trajectory experiment of stratosphere tracers?

Thank you for this suggestion. Given the computational cost of high-resolution model output, we calculated the 5-day forward trajectory of stratospheric air using the NOAA HYSPLIT Model. The location of trajectory was set to Binzhou (37.7 $^{\circ}$N, 118.1$^{\circ}$E, the black star symbol in the figure) at an altitude of 10 m at 04:00 LST on 1 August 2021. The forward trajectory from HYSPLIT shows that the stratospheric air that had reached the ground remained in the planetary boundary layer (<1500 m) for the next five days, suggesting the important consequences of such an ozone enhancement with stratospheric origin.

**NOAA HYSPLIT MODEL**
**Forward trajectory starting at 2000 UTC 31 Jul 21**
**GFSQ Meteorological Data**

[Figure]

Job ID: 164681          Job Start: Tue May 10 08:21:50 UTC 2022
Source 1    lat.: 37.700000    lon.: 118.100000    height: 10 m AGL

Trajectory Direction: Forward      Duration: 120 hrs
Vertical Motion Calculation Method:      Model Vertical Velocity
Meteorology: 0000Z 31 Jul 2021 - GFS0p25

---

## Author Comment (AC5)

**#Reviewer 5**

**General comments:**

This is an interesting paper, addressing an important topic, and appropriate for publication in ACP. At times the description of the dynamics is hard to follow, and so I suggest some attention be given to making the arguments simple and clear.

The paper mainly falls down, I think, in presenting the meteorology of the convective system that is alleged to be responsible for bringing the ozone down to the surface. This may simply be because the authors are meteorologists by training, and forget that most ACP readers are not. Please explain more! A prime example is Figure 11, which purports to demonstrate the downward transport: "We performed cross-section analyses of the bow-echo MCS (Fig. 11b-c), and the results clearly show a rearward pathway through which the stratospheric ozone-rich airmass was transported to the surface by the rear inflows descending from stratiform clouds to the leading convective line." Maybe they clearly show that to the authors, but unfortunately not to this reader. Is the salient point that the "tracers" are now mostly below 3.6 km? Or that some of the wind vectors are pointing down? Is the reader supposed to be able to see "the rear inflows descending from stratiform clouds to the leading convective line"? Remember that many of your readers won't know where the "leading convective line" is found.

Thank you for your points. We have tried out best to add more related information in the manuscript and modified the expressions. The point-to-point responses are listed below.

**Minor points:**

The presentation and grammar need some good editing. It is sometimes difficult to understand what the authors are trying to say.

Sorry for this, we have tried out best to improve the language of this paper. Many thanks to the reviewers for pointing out the grammar errors.

Abstract: This is a bit long, and some of it reads like an introduction to the paper, rather than a brief summary of new results. At the least, the last sentences of each paragraph (lines 20-21 and 30-32) should be moved or deleted.

Thank you, we have modified the sentences and shortened the abstract.

Lines 66, 449, 450: "…wrapped around the anvil". An "anvil" is a block of iron that a blacksmith hammers upon. "Rearward anvil" and "forward anvil" are meteorologist's slang. Most readers will know that an anvil-shaped cloud is often associated with a thunderstorm, but no more. Please be clear about what you are describing, and why.

The anvils form and extend in both directions because the updrafts of thunderstorm divide into two branches at upper levels. Readers are referred to Houze (2004) for more details about the anvil clouds and MCSs.

Houze Jr., R. A.: Mesoscale convective systems, Rev. Geophys., 42, RG4003, https://doi.org/10.1029/2004RG000150, 2004.

Lines 111-116: The instrumentation should be identified, and/or the uncertainty and detection limits cited.

The ground-based air pollutant data from CNEMC have been widely used in researches concerning air quality and air pollution in China. Readers are referred to Lu et al. (2018), Li et al. (2019, 2020), Han et al. (2020), Wang and Zhang (2020) for more details about the observation network.

Lu, X., Hong, J., Zhang, L., Cooper, O. R., Schultz, M. G., Xu, X., Wang, T., Gao, M., Zhao, Y., and Zhang, Y.: Severe surface ozone pollution in China: A global perspective, Environ. Sci. Technol. Lett., 5, 487–494, https://doi.org/10.1021/acs.estlett.8b00366, 2018.

Li, K., Jacob, D. J., Liao, H., Shen, L., Zhang, Q., and Bates, K. H.: Anthropogenic drivers of 2013–2017 trends in summer surface ozone in China, P. Natl. Acad. Sci. USA, 116, 422, https://doi.org/10.1073/pnas.1812168116, 2019.

Li, K., Jacob, D. J., Shen, L., Lu, X., De Smedt, I., and Liao, H.: Increases in surface ozone pollution in China from 2013 to 2019: anthropogenic and meteorological influences, Atmos. Chem. Phys., 20, 11423–11433, https://doi.org/10.5194/acp-20-11423-2020, 2020.

Han, H., Liu, J., Shu, L., Wang, T., and Yuan, H.: Local and synoptic meteorological influences on daily variability in summertime surface ozone in eastern China, Atmos. Chem. Phys., 20, 203–222, https://doi.org/10.5194/acp-20-203-2020, 2020.

Wang, X. and Zhang, R.: Effects of atmospheric circulations on the interannual variation in PM2.5 concentrations over the Beijing–Tianjin–Hebei region in 2013–2018, Atmos. Chem. Phys., 20, 7667–7682, https://doi.org/10.5194/acp-20-7667-2020, 2020.

Lines 189-190:    I think this is saying that FLEXPART-WRF used the 3-km resolution output of WRF-ARW, but it isn't really clear.

Sorry for this. We have modified the sentence in the manuscript.

Lines 208-201: "It is a common practice to use 25th percentile of ozone concentration distributions as a background value (e.g., Parrington et al., 2013), which yields an even severer ozone enhancement in the surface."    I think the authors are trying to suggest that the ozone amount is more significant because it is all transported from elsewhere, and so could be measured against some "background" value (arbitrarily defined). This is a dubious comparison that will only serve to confuse the reader. Delete.

Thank you for your suggestion, this sentence has been deleted.

Line 226: After "…not reduced in Qingdao and Weihai" I suggest adding "…which were outside of the path of influence of the MCS, as noted in the preceding paragraph."

Thank you. Added in the manuscript.

Line 230: I suggest referring to Figure 1 here.

Thank you for this suggestion. Added in the manuscript.

Line 281: Bohai Bay is not indicated in Figure 1.

Sorry for this. We changed it to "Bohai Sea" in the manuscript.

Lines 287-288: Is that the blue areas? In other words, does a positive vertical velocity in Pa imply downward motion? This is not clear.

Yes, the vertical velocity here uses a unit of Pa $s^{-1}$, and hence the positive values (in blue colors) represent the downward air motions. We added more description in the caption of Fig. 5

Line 307, Figure 6: Why use dewpoint depression? This metric will be unfamiliar to those without meteorological training (most ACP readers!).

Sorry for this. The temperature and dewpoint are frequently used in the radiosonde observations, and the differences between them (dewpoint depression) directly imply the saturation of airmass. For this reason, we compared the dewpoint depressions to track the descent of stratospheric dry air.

Line 344: The term "bow-echo MCS" is used here without definition. The description appears later, beginning on line 362. Please move it ahead of this.

Thank you for this valuable suggestion. The description of bow-echo MCSs was moved to Line 356-360.

Line 363: "produce"? Perhaps "are associated with" would be better. The radar echoes don't cause the winds.

Thank you. Modified in the manuscript.

Line 367: Perhaps they should be shown? I find the evidence of descent unconvincing at present, and this is an important part of the paper. Also, a few lines below, you claim that "…strong radar reflectivities were confined below 6 km altitude (480 hPa, -9 ℃) suggesting limited vertical extension of convective storms." It would be helpful to see those data.

The figure below shows the radar reflectivity and radial velocity observed by Jinan Radar station (white cross symbol) at 01:00 LST on 1 August 2021 with a radar elevation angle of 0.5°. The storm had moved close to the radar station (figure a). In the radial velocity filed (figure b), the negative values represent that the airmasses move toward the radar station (white cross symbol here) and the positive values represent the airmasses move away form the radar. A region associated with large negative radial velocities exceeding 20 m/s (the magenta dashed circle) appeared in the north of radar station due to the development of descending rear inflows of bow echoes.

[Figure]

"the strong radar reflectivities were confined below 6 km altitude (480 hPa, -9 °C) suggesting limited vertical extension of convective storms.". The supporting data and figure is shown in Fig. S8 in the supplementary file.

Lines 388-394: This plot and description give me no useful information with which to evaluate the model performance. What exactly is being simulated? What observations are being compared? Does a POD of 0.8 mean we have 80% perfect agreement, or 80% chance of seeing something similar within 20 km? What does the SR of 20-80% mean, and what is a frequency bias (FR)?

In contrast, I do get some information from comparing Figures 8 and 9. Perhaps instead of S9 you could simply describe the agreement between these figures. It looks to me like WRF is simulating a system of similar size and strength in pretty much the same place.

Yes, we can qualitatively evaluate the model performances by comparing Figures 8 and Figure 9. The WRF model does a good job capturing the evolution of the MCSs. Additionally, we intended to quantitatively compare the model simulation and observations in the form of performance diagram, which is frequently used in high-resolution weather models. The simulated radar reflectivity is compared with the observed one with the following performance metrics. Success ratio (SR), which equals 1-Far (False Alarm Rate), POD (probability of detection), CSI (critical success index) and FB (frequency bias). According to the contingency table (Table S1), there are hits, misses, false alarms, and correct negatives.

Table Contingency table for observations and simulations

|  |  | Observations | |
|  |  | Yes | No |
| --- | --- | --- | --- |
| Simulations | Yes | Hits | False alarms |
|  | No | Misses | Correct negatives |

And hence we calculated the following metrics with a 20-km searching radius.

$POD = \frac{Hits}{Hits+Misses}$ , a ratio that ranges from 0 to 1 and represents the ratio of correctly forecasted objects to the total number of observed objects.

$FAR = \frac{False\ alarms}{False\ alarms\ +Correct\ negatives}$ , a ratio ranges from 0 to 1 and represent the ratio of overproduced forecasts to the total number of objects forecasted, which also ranges from 0 to 1. Thus SR (= 1-FAR) represent the ratio of correctly forecasted objects to the total number of objects forecasted.

$CSI = \frac{Hits}{Hits+False\ alarms\ +Misses}$ , a ratio of correctly forecasted objects to the total number of observed and forecasted objects

$FB = \frac{Hits+False\ alarms}{Hits+Misses}$ , a ratio of the positive forecasts (both true and false) to the number of observed objects.

Lines 442-448: This description and Figure 11 are quite confusing to me, as noted above. I'm not at all sure what the lines labelled "tracers" represent. Are they contours of particle counts? At 3.6 km?

The magenta contour lines labelled "tracers" represent the number of tracer particles at each altitude. We have modified the caption in Fig. 11.

Figures 2, 3 & 4: I find the times on the X-axis hard to read.

The times on the X-axis are expressed as month/day/hour, and we have added more description in the figure caption.